# Exposure to the Russian Internet Research Agency foreign influence campaign on Twitter in the 2016 US election and its relationship to attitudes and voting behavior

Gregory Eady [1] ✉, Tom Paskhalis [2] ✉, Jan Zilinsky [3], Richard Bonneau [4], Jonathan Nagler [5] & Joshua A. Tucker [5]

There is widespread concern that foreign actors are using social media to interfere in elections worldwide. Yet data have been unavailable to investigate links between exposure to foreign influence campaigns and political behavior. Using longitudinal survey data from US respondents linked to their Twitter feeds, we quantify the relationship between exposure to the Russian foreign influence campaign and attitudes and voting behavior in the 2016 US election. We demonstrate, first, that exposure to Russian disinformation accounts was heavily concentrated: only 1% of users accounted for 70% of exposures. Second, exposure was concentrated among users who strongly identified as Republicans. Third, exposure to the Russian influence campaign was eclipsed by content from domestic news media and politicians. Finally, we find no evidence of a meaningful relationship between exposure to the Russian foreign influence campaign and changes in attitudes, polarization, or voting behavior. The results have implications for understanding the limits of election interference campaigns on social media.

When the major social media platforms first emerged in the mid-2000s, they were credited with providing essential collective action tools for democratic activists and with spurring several high-profile social movements worldwide (e.g., Iran's Green Wave movement, the Arab Spring, Occupy Wall Street)[1]. Yet the initial optimism that surrounded the democratizing potential of social media was short-lived. Governments soon recognized the collective action potential of social media, and responded by developing strategies to use these platforms for their own domestic and foreign policy goals. A number of high-profile cases have since suggested that some governments are using social media to undermine the social movements that challenge their domestic power, and to intervene in the democratic elections of their foreign adversaries[2]. An example of such a foreign intervention was the one conducted during the 2016 US election campaign by the Internet Research Agency, an organization closely linked to the Russian government[3]. The Internet Research Agency's alleged efforts to undermine US democracy are now widely documented by the news media[4], US government investigators[3,5], and researchers[6-8]. The organization is accused of using social media accounts impersonating US users to polarize the US electorate and influence the attitudes and voting behavior of ordinary Americans during the 2016 US election campaign. Social media companies once again warned that Russia-based organizations were seeking to intervene in the 2020 US presidential election[9].

[1]Department of Political Science and the Center for Social Data Science (SODAS), University of Copenhagen, Copenhagen, Denmark. [2]Department of Political Science, Trinity College Dublin, Dublin, Ireland. [3]Department of Governance, Technical University of Munich, Munich, Germany. [4]Department of Biology and Computer Science and Center for Social Media and Politics, New York University, New York, NY, USA. [5]Department of Politics and Center for Social Media and Politics, New York University, New York, NY, USA. ✉e-mail: gregory.eady@gmail.com; tpaskhalis@gmail.com

Foreign influence campaigns have attracted substantial popular and academic interest[10–12]. Researchers' understanding of the influence of the social media side of these campaigns remains unresolved, however, in large part due to the absence of the data. Previous research has, for example, examined the relationship between interactions with Russian foreign influence accounts in the United States and attitudes and political behavior[12]. Yet the authors acknowledge a number of limitations: the available data are from a year after the 2016 US election occurred, cover a short one-month time window, and were collected after Twitter removed many Russian foreign influence accounts from its platform. Other research on foreign influence campaigns has sought to understand the structure and content of these campaigns[6–8,13,14], but has not assessed the relationship between exposure to content from foreign influence accounts and political attitudes, polarization, and vote choice.

In this article, we investigate the relationship between Russia's foreign influence campaign on Twitter during the 2016 US presidential election and the political attitudes and voting behavior of ordinary US social media users. We link longitudinal survey data from a sample of US Twitter users with data from those respondents' social media feeds that were collected during the 2016 campaign. These survey-linked social media data allow us to both quantify the distribution and scale of ordinary US users' exposure to posts from Russian foreign influence accounts, and to estimate the relationship between exposure to these accounts and users' positions on policy issues, political polarization, and voting behavior in the 2016 election.

Theoretically, there are good reasons to expect both why foreign influence campaigns on social media might succeed and why they might fail. One reason to expect that foreign influence campaigns could affect the attitudes and behaviors of social media users is their seemingly large scale and reach. We estimate, for example, that at least 32 million US Twitter users were potentially exposed to posts from Russia-sponsored accounts in the eight months leading up to the 2016 election. Facebook has estimated, by comparison, that 126 million users had the potential to view Russian state-sponsored content on its platform over a two year period[15]. As Facebook in 2016 was used by roughly 3.5 times as many Americans as Twitter, this suggests that the reach of Russian foreign influence campaign content across both platforms was similar. Moreover, researchers and government investigators are consistent in their assessment of the potential goals of the Russian foreign influence campaign[6–8,10]. First, they generally agree that Russian interference in the US election was designed to influence the voting behavior of US users in favor of Donald Trump, either by shifting support toward Trump himself, or by encouraging disaffected liberals—often former Bernie Sanders voters—to vote for a third-party candidate or to abstain from voting altogether. Second, US government reports conclude that the Russian foreign influence campaign sought to undermine US democracy more generally by exacerbating polarization among the electorate[3,16].

Yet, despite the Russian foreign influence campaign's apparent scale and intentions, one should be skeptical about its potential effects on attitudes and voting behavior. The large body of political science research that examines the effects of traditional election campaigns on voting behavior finds little evidence of anything but minimal effects[17,18], even when messages are well-targeted and conducted in politically conducive environments[19,20]. Furthermore, although the scale of the Russian foreign influence campaign is seemingly impressive in absolute terms (i.e., millions of exposures), its scale may pale in relative terms to social media users' exposure to other political content. Absent a benchmark against which to measure the scale of US users' exposure to posts from foreign actors, however, it remains difficult to assess the potential relationship between exposure to that content and changes in political attitudes and voting behavior. Finally, although the alleged intention of the Russian foreign influence campaign on social media was to influence the attitudes and behavior of voters in ways favorable to Donald Trump, the extent to which exposure was concentrated among a small number of users, or those most or least likely to be affected, is unknown. Recent research on fake news, for example, shows that exposure to fake news content is concentrated among a small group of users, and those who identify as strong political partisans[21]. If exposure to social media posts from Russian foreign influence accounts during the 2016 US election was similarly concentrated (i.e., among those already favorable toward Donald Trump), then their influence on changing candidate preferences could be minimal.

We adjudicate this debate by examining the relationships between exposure to posts from the Russian foreign influence campaign and the political attitudes and behaviors of ordinary US social media users. We triangulate two types of evidence: (1) descriptive evidence to investigate the relative scale and concentration of US users' exposure to posts from Russian foreign influence accounts, and the partisan profile of those exposed, and (2) evidence from social media-linked longitudinal survey data that we use to examine the relationship between exposure to posts from Russian foreign influence accounts and changes in respondents' political attitudes and voting behavior during the election campaign.

Here, we demonstrate the extent to which exposure to posts from foreign influence campaigns was concentrated among a small group of users. We show, further, that although the amount of exposure to social media posts from foreign influence campaigns is large in absolute terms, it was overshadowed—by at least an order of magnitude—by content from ordinary domestic political news media and US political candidates. Finally, we examine whether exposure to posts from the Russian foreign influence campaign is associated with changes in US respondents' positions on salient election issues, levels of political polarization, and voting behavior in the 2016 election. Across a wide range of outcomes, we do not detect a meaningful relationship between exposure to posts from Russian foreign influence accounts and changes in attitudes, polarization, or voting behavior.

## Results

### Exposure to Russian foreign influence accounts during the 2016 election campaign

We begin by answering key questions regarding the levels and concentration of US social media users' exposure to posts from foreign influence accounts during the 2016 election. We do this by examining potential exposure (hereafter "exposure" - see Methods section for details) to posts from Internet Research Agency accounts and those from smaller foreign influence campaigns (from China, Iran, and Venezuela) for comparison. We find that 70% ($n = 1042$) of respondents were exposed to one or more posts from a foreign influence campaign between April and November 2016, with 786,634 posts from these campaigns identified across all respondents' timelines ($n = 1496$). To examine variation in exposure over time, we present in panel a of Fig. 1 the average levels of exposure to posts from foreign influence campaigns each day in the eight months prior to the election. As the figure shows, the daily volume of exposure to tweets from foreign influence campaigns, aggregated over all respondents' timelines, varied widely between roughly two thousand early in the election campaign to roughly ten thousand at its height, with a peak on election day of roughly 24,000 exposures. Individual respondents were exposed on average to between two and ten posts from foreign influence campaigns per day. As panel a of Fig. 1 shows further, posts from the Russian influence campaign were the most prevalent, representing 86% of all exposures in respondents' timelines in the lead-up to the election among the foreign influence campaigns included herein.

The main avenue for exposure to these posts was not through users directly following foreign influence accounts. Exposure was mostly incidental, primarily via retweets from ordinary accounts that users followed. The share of retweets as an exposure pathway grew

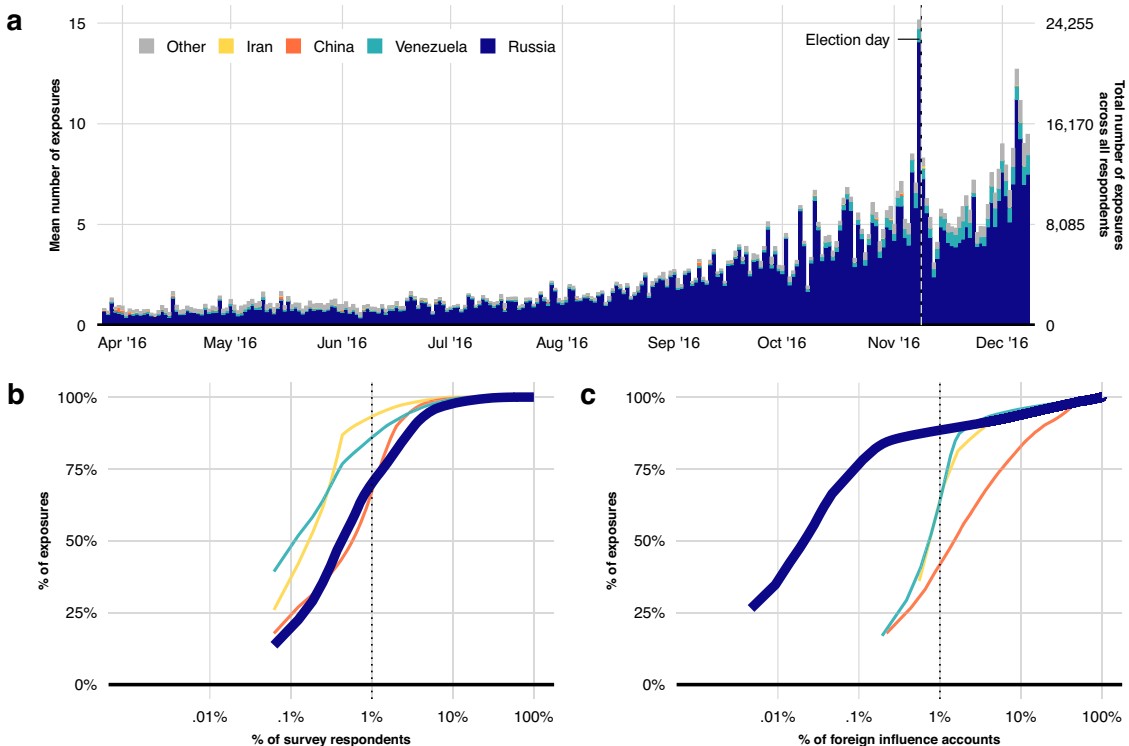

**Fig. 1 | Exposure to tweets from state-sponsored accounts over time among survey respondents.** Panel **a** presents the total number of exposures to tweets from foreign influence campaigns during the 2016 US election campaign among respondents between the first survey wave, and one month after the election. Panel **b** presents the empirical cumulative distribution of exposure to tweets by survey respondents from Russian, Venezuelan, Chinese, and Iranian foreign influence accounts. Panel **c** presents the empirical cumulative distribution of the tweets sent by accounts from each foreign influence campaign. The line representing the cumulative distribution for Russian influence campaign accounts begins visually at a different point on the x-axis due to the higher number of Russian foreign influence accounts relative to other smaller state-backed campaigns. Vertical dotted lines in **b**, **c** highlight the percentage of exposures by 1% of respondents (panel **b**), and 1% of state-sponsored accounts (panel **c**).

steadily over time, reaching 75–80% by election day (see Supplementary Fig. C4). However, despite the seemingly large number of exposures to posts from foreign influence campaigns, the amount of exposure is meaningful only within the context of the political ecosystem on social media. This is because social media users are exposed to a wide array of posts concerning politics from legitimate political actors every day, especially during a high-profile election campaign. To put the number of posts from the Russian foreign influence campaign in perspective, we identify the posts in respondents' timelines that were from US national news media organizations, and politicians and candidates, during the last month of the election campaign, when Internet Research Agency activity was at its highest point. For comparison, we present the mean number of posts our respondents were exposed to by the news media, politicians, and Internet Research Agency accounts side-by-side in panel a of Fig. 2. Despite the seemingly large number of posts from Internet Research Agency accounts in respondents' timelines, they are overshadowed—by an order of magnitude—by posts from national news media and politicians. While, on average, respondents were exposed to roughly 4 posts from Russian foreign influence accounts per day in the last month of the election campaign, they were exposed to an average of 106 posts on average per day from national news media and 35 posts per day from US politicians. In other words, respondents were exposed to 25 times more posts from national news media and 9 times as many posts from politicians than those from Russian foreign influence accounts. This contrast is similarly large in panel b of Fig. 2, which presents median exposure to posts from Russian foreign influence accounts per week. In the last month of the election, the median exposure to Russian foreign influence accounts is zero across all weeks, because, as we show below, exposure is concentrated among a small group of users,

and thus there are few who are exposed to any posts from foreign influence accounts at all in a given week.

That median exposure is zero per week in the last month of the election suggests that exposure may, in general, be concentrated among a small group of users. To examine this, we present in panel b of Fig. 1 the cumulative distribution of exposure among respondents to posts from foreign influence accounts. It shows that exposure to foreign influence accounts is concentrated among a small group of respondents: 1% of respondents account for 70% of exposures to posts from Russian foreign influence accounts. Furthermore, almost all exposure is concentrated among only 10% of respondents, who account for 98% of exposures to posts from Russian foreign influence accounts. Interestingly, this concentration of exposure among a small group of users is broadly similar to patterns found in studies of exposure to fake news[21,22]. It is also worth pointing out that this is not a feature of political communication on Twitter per se. Analogous concentration plots for politicians and news media in Supplementary Methods D show that 1% of respondents account for 24% and 37% of exposures to posts from domestic news media and politicians, respectively. In other words, exposure to Russia foreign influence accounts is particularly concentrated among a small subset of users. Finally, we also show in panel c of Fig. 1 that a small numbers of Russian foreign influence accounts are responsible for a large majority of these exposures, with 1% of Russian accounts accounting for 89% of the content found in individuals' timelines.

## Predictors of exposure to the Russian foreign influence campaign

We now examine the characteristics of users who were more likely to be exposed to Twitter posts from the Russian foreign influence campaign,

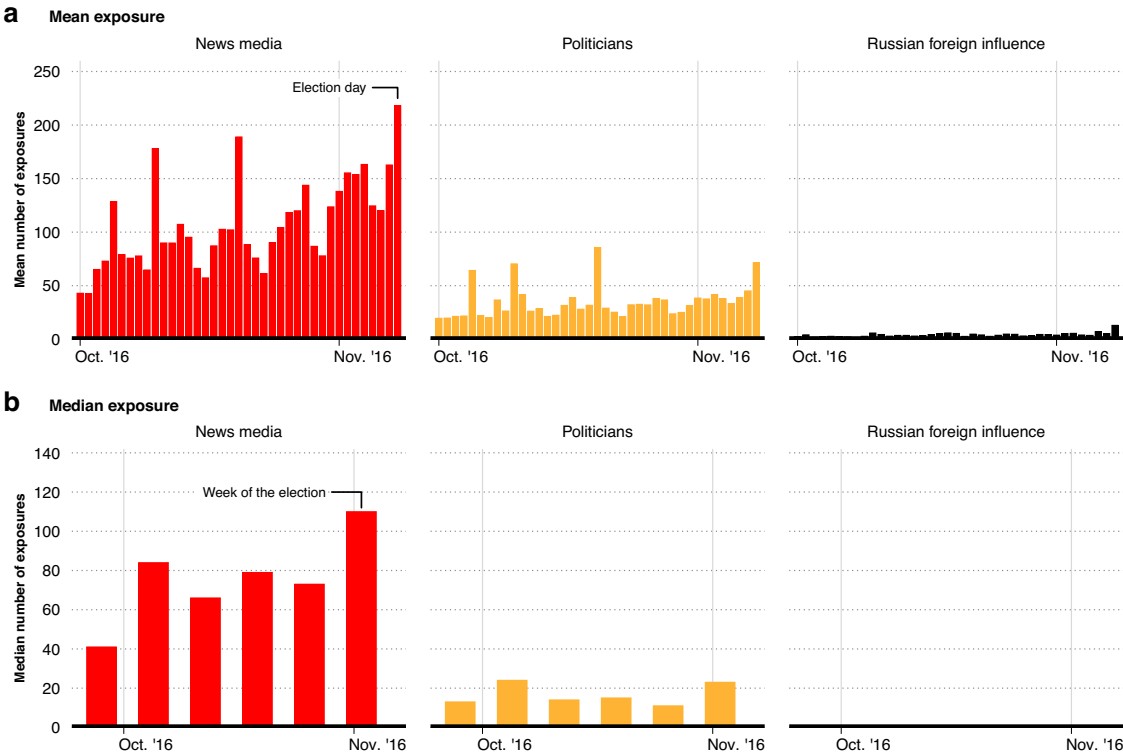

**Fig. 2 | Average exposure to posts from domestic news media, politicians, and Russian influence campaign accounts during the final month of the election campaign.** Panel **a** presents the mean exposure to posts from domestic news media (in red), political candidates (in orange), and Russian foreign influence accounts (in black) for the last month of the presidential campaign. Panel **b** presents an analogous time series per week for median exposure to domestic news media, political candidates, and Russian foreign influence accounts. Median exposure to Russian foreign influence accounts in the last month of the election is 0 for all days.

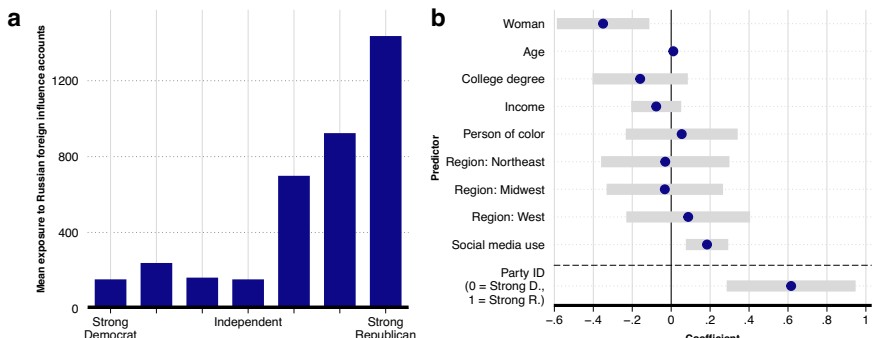

**Fig. 3 | Partisanship and exposure to posts from Russian foreign influence accounts during the 2016 US election campaign.** Panel **a** presents the mean exposure to posts from Russian foreign influence accounts among survey respondents according to their self-identified partisanship. Panel **b** presents the coefficients (and 95% CIs) from an OLS model where the outcome variable is the (log exposures + 1) number of tweets from Russian foreign influence accounts in a respondents' Twitter timeline between March 25 and November 8, 2016. The model intercept parameter is not shown. The reference category for the region variable in the model is "South". $n = 1496$ survey respondents.

with particular attention to differences in political partisanship. If the Internet Research Agency's alleged efforts were thought of as a typical political campaign, we would expect that those efforts would take one of three forms: persuading voters who might not support the campaign's preferred candidate to do so; mobilizing a favored candidates' supporters; or demobilizing the supporters of an opponent. These strategies would predict that tweets from Russian foreign influence accounts would, in general, be aimed primarily at those who identify as moderates, Republicans, or Democrats respectively. Of course, the Internet Research Agency could also have employed multiple strategies, especially if an additional goal were to increase political polarization.

To examine this, we begin by presenting in panel a of Fig. 3 the average exposure to posts from Russian foreign influence accounts over the course of the campaign broken down by how survey respondents identify their partisanship. Results in panel a show that the amount of exposure depends substantially on users' self-identified partisanship: those who identify as "Strong Republicans" were exposed to roughly nine times as many posts from Russian foreign influence accounts than were those who identify as Democrats or Independents. To examine whether the findings in panel a are an artifact of factors other than partisanship, we fit an OLS regression model to predict the (log exposures + 1) number of posts from Russian foreign influence

accounts that were in the timelines of each respondent (for results from a Poisson model, see Supplementary Methods C3). Including a set of standard socio-demographic characteristics, a control for respondents' level of social media use, and a 7-category party ID variable as predictors allow us to verify that the simple party-exposure correlation is not driven by demographics specific to survey respondents who identify as Republicans. Results from this model are presented in panel b of Fig. 3. Consistent with results in panel a, the more strongly a user identified as a Republican, the more posts in their timeline from Russian foreign influence accounts.

We investigate this result further by examining whether the amount of exposure to the Russian foreign influence campaign was not only greater among those who identify as highly partisan Republicans, but also among those who identify as highly partisan Democrats. If the relationship were U-shaped in this way, such that those who identify as highly partisan Democrats and Republicans were similarly exposed, it would suggest another potential pathway for a relationship between exposure to Russian foreign influence accounts and political attitudes and voting behavior. For example, exposure among those who identify as highly partisan Democrats could plausibly lead to increases in disaffection with the more moderate Democratic nominee, thereby encouraging voting for a third-party candidate or abstaining from voting altogether. To test this, we include a squared party identification term in the regression model. As we show in the Supplementary Methods C2, we do not find statistical evidence that exposure to posts from Russian foreign influence accounts was as high among those who identify as highly partisan Democrats as it clearly was among those who identify as Republicans: inclusion of the squared term does not significantly increase model fit, and as we show in Supplementary Methods C2, the relationship between party ID and exposure is monotonic across the range of the party identification variable. We find, in other words, that exposure to Russian foreign influence accounts was concentrated among those who identify as highly partisan Republicans—those most likely to already strongly support the Republican nominee. Exposure was not, however, similarly concentrated among those who identify as highly partisan Democrats.

### Aggregate US Exposure to Russian foreign influence accounts on Twitter during the 2016 election campaign

In addition to examining the individual-level characteristics of individuals exposed to Russian foreign influence accounts, we also calculate a rough estimate of the aggregate exposure across the United States. While as a result of congressional hearings on the role of social media in the 2016 US presidential election, both Twitter and Facebook calculated and released aggregate-level estimates of user exposure to Russian foreign influence accounts on their platforms, each company did so differently, with statistics that are not comparable. Facebook, for example, stated that 126 million of its users had the potential to view content from the Russian foreign influence campaign over a two year period[15]. Twitter, however, states that the number of times that content from Russian foreign influence accounts was viewed within a brief two and a half months (September 1, 2016 to November 15, 2016) was 288 million[23]. This is likely a large underestimate of the number of views, due to the fact that Twitter later expanded its list of Russian foreign influence accounts[24], but without providing an updated number of views. With 1.4 million direct interactions with that content (e.g., liking, retweeting, replying)[24], the figure 288 million views, however, is an extreme upper bound on the number of potentially exposed users, because a single user can view multiple tweets from Russian foreign influence accounts. By contrast, 1.4 million interactions is an extreme lower bound on potential exposure, because the vast majority of users do not directly interact with each piece of content on the platform.

Thus, to aid our understanding of the overall potential exposure of US Twitter users, we make a back-of-the-envelope calculation of the number of US users potentially exposed to posts from Russian foreign influence accounts during the 8 months leading up to election day. To do so, we use data on potential exposure to Internet Research Agency posts among users in our dataset, which we combine with data from Pew Research and US census data. Pew Research estimates that in 2016, Twitter penetration in the US was 21%[25]. The US Census Bureau estimates that the US population in 2016, aged 18 and older, was 244,807,000[26]. This puts a rough estimate of the number of American Twitter users at 51 million. Taking the estimate from our social media-linked survey data that 63% of US Twitter users were potentially exposed to at least one post from Russian foreign influence accounts during the 2016 presidential election (see Supplementary Methods C1), a back-of-the-envelope calculation (0.63 × 51 million) suggests that 32 million Americans were potentially exposed to content from Internet Research Agency accounts. Note that this figure is not perfectly analogous to Facebook's to the extent that their estimate is across 2 years prior the election.

### The relationship between exposure to Russian foreign influence accounts and political attitudes and polarization

We now document the relationship between exposure to posts from the Russian foreign influence campaign and changes in respondents' (1) positions on salient election issues and (2) perceptions of candidate polarization. We measure political attitudes by the positions that respondents took in the survey on eight major policy issues that were salient during the election, and respondents' self-reported political ideology. Examples of policy issues in the survey include respondents' opinions toward the expansion of the Affordable Care Act, increases in tariffs on China, building a wall on the border with Mexico, and Donald Trump's call to ban Muslim people from traveling to the US. Positions on each issue were measured on a 0 to 100 scale, labeled with end points indicating the direction of support (for survey text for each issue question, see Supplementary Methods A3). Because our interest is in the relationship between exposure to Russian foreign influence accounts and changes in issue positions as they relate to the presidential candidates, we recode the scales so that higher values indicate closer alignment with Trump (e.g., approval of building a wall on the border with Mexico), and lower values indicate closer alignment with Clinton (e.g., disapproval of a wall). Survey respondents also placed Trump and Clinton on the same eight issue scales and ideological scale. These candidate placements allow us to capture perceived polarization, measured as the distance between these placements such that higher values indicate higher polarization, i.e., a belief that the candidates are further apart on the issues and ideologically (for details see Supplementary Methods F1).

To investigate the relationship between exposure to Russian foreign influence accounts and political attitudes we take advantage of the panel structure of the data. These data allow us to examine within-subject variation, and therefore to account for time-invariant characteristics across respondents. We model the relationship between exposure and issue positions and ideology by regressing within-respondent changes in issue positions and perceptions of polarization on exposure to posts from Russian foreign influence accounts between survey waves. In other words, we examine whether exposure is associated with changes in each respondent's political attitudes and perceived polarization from before the campaign to immediately prior to the election. If exposure were unconfounded between survey waves (a strong assumption), the estimand in this model would be the average treatment effect on the treated to the extent that levels of exposure is as observed in the data. The regression model is also equivalent to one that predicts a respondent's attitude toward an issue in the final wave of the survey conditional on the position that they took on that issue in the first wave of the survey (with a coefficient of 1) and their exposure to posts from Russian foreign influence accounts. Because exposure to foreign influence accounts is concentrated among a relatively small group of users (as shown in panel b of Fig. 1), we note that our estimates

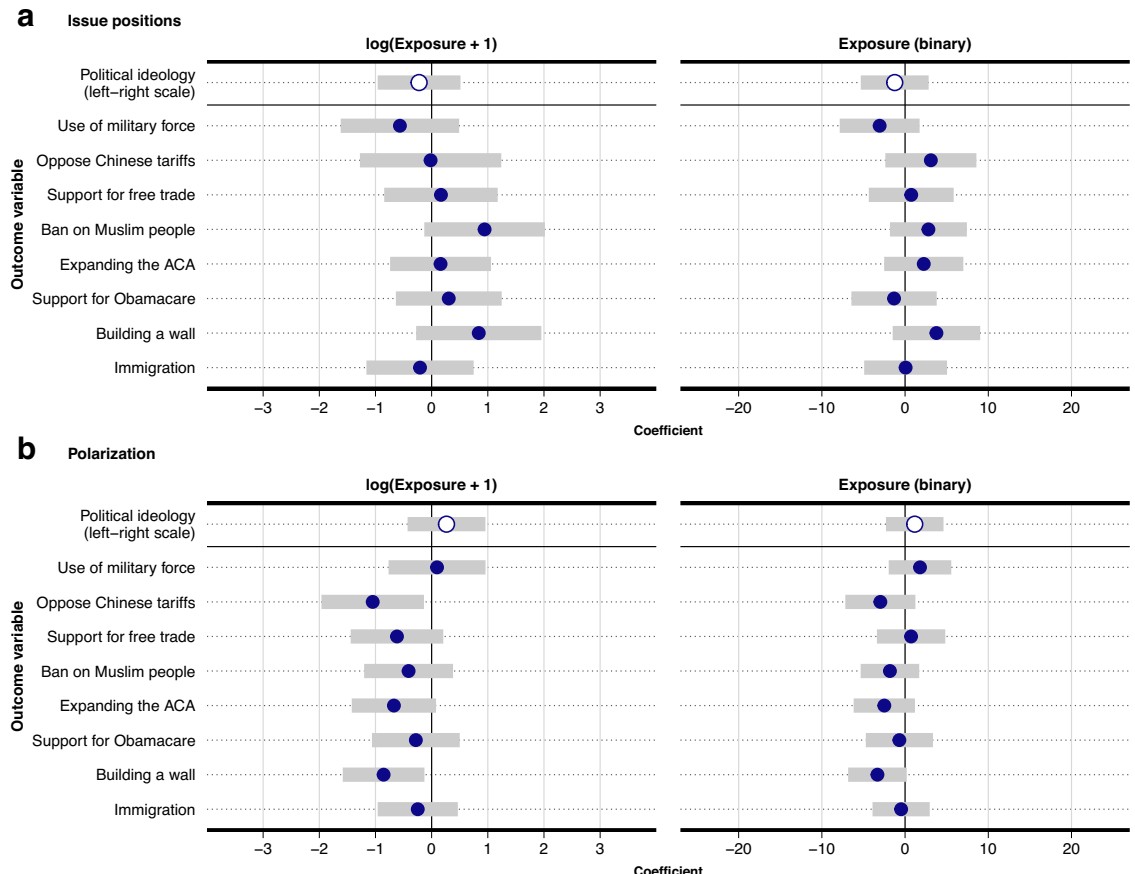

**Fig. 4 | Regression results of exposure to Russian foreign influence accounts and changes in issue-based and ideological positioning.** Each point (with 95% CIs) represents an OLS-estimated association (each from a separate model) between exposure to posts from Russian foreign influence accounts and political ideology, and eight issue-based outcomes that are taken from questions asked in each wave of the survey. Two types of models are presented, with exposure coded first as the (log exposures +1) number of tweets from Russian foreign influence accounts appearing in a respondents' Twitter feed (left column), and second, whether at least one Russian tweet appeared in a respondents' feed (right column).

Each outcome is measured as the within-respondent change in positioning between the first and third wave of the survey. Panel **a** shows changes in the relative issue positions of respondents, such that movement in the positive direction represents one favorable to Donald Trump. Panel **b** shows changes in perceived polarization between Donald Trump and Hillary Clinton on each issue, such that positive values indicate increases in perceived polarization during the election campaign, and negative values indicate decreases in perceived polarization. $n = 1496$ survey participants.

in the following sections are driven by those (heavily) exposed to messages from these accounts. Were exposure distributed differently, among another set of users, the estimated relationship could well be different. It should thus be kept in mind that those exposed to foreign influence accounts were users who self-identified as highly partisan Republicans, a fact that in itself aids in contextualizing the limited scope of the Russian foreign influence campaign.

The main results are presented in Fig. 4 (see Supplementary Methods F for complete output). As the figure shows, we do not find statistical evidence in support of a relationship between exposure to posts from Russian foreign influence accounts and changes in respondents' issue positions or perceptions of polarization. This is the case regardless of whether exposure is measured as the (log exposures +1) number of exposures to posts from Russian foreign influence accounts or as a binary variable indicating whether a respondent was exposed to at least one such post. For ideological and issue positions, the estimated relationships are neither significant nor in a direction consistent with one favorable toward Donald Trump. This result is similar for polarization. Of the two statistically significant coefficients across all models (representing 6% of our coefficients, as would be expected by chance), neither is in a direction that would suggest that exposure to posts from Russian foreign influence accounts is related to an increase in perceived

polarization. Finally, adjusting for multiple comparisons (see Supplementary Methods G), we do not find statistically significant relationships between exposure on any of the issue-based and polarization outcomes.

Because the absence of statistically significant results is not necessarily strong evidence of a negligible relationship, we use equivalence testing to place bounds on the magnitude of each relationship that can be statistically rejected[27–29]. We calculate bounds for each estimate shown in Fig. 4 using Two One-Sided Tests (TOST) (see Supplementary Methods B). The magnitudes of the estimated relationships in standardized units are near zero: 0.05 standard deviations for changes in issue positions; 0.06 standard deviations for changes in perceived polarization. For all but one of the 18 outcomes, equivalence testing demonstrates that the relationships are not >0.2 SDs, i.e., rejecting the hypothesis that any relationship is >0.2 SDs of the issue outcome variable ($p < 0.05$ for all but one outcome) (see Supplementary Methods B for details). For comparison, this inability to detect meaningful relationships between exposure and changes in issue positions is consistent with recent large-scale field experimental research in the US that finds near-zero effects ($\beta \approx 0.01$ SD) of exposure to targeted issue advertisements on changes in issue positions (LGBTQ and immigration policy preferences)[30].

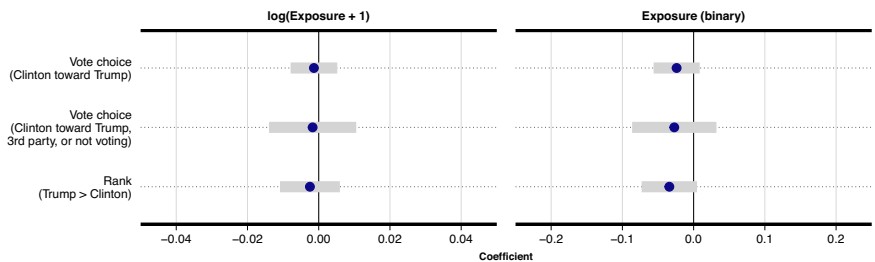

**Fig. 5 | Regression results of the relationship between exposure to posts from Russian foreign influence accounts and voting behavior.** These panels present OLS estimates (with 95% CIs) of the relationship between exposure to tweets from Russian foreign influence accounts and three vote choice outcomes. Each estimate is from a separate model (see Supplementary Methods E for full regression results). In the first row, the outcome "Vote choice (Clinton toward Trump)," is coded with three possible values: (+1) a shift from ranking Clinton preferable to Trump in the first wave to voting for Trump in the election; (−1) a shift from ranking Trump preferable to Clinton to voting for Clinton; or (0) no difference between first-wave ranked preferences and vote choice in the election. The second row outcome variable, "Vote choice (Clinton toward Trump, 3rd party, or not voting)" can also take 3 values: (+1) a shift from ranking Clinton above Trump in the first survey way to voting for Trump, voting for a 3rd party candidate, or not voting; (−1) a shift from ranking Trump above Clinton to voting for Clinton, voting for a 3rd party candidate, or not voting; or (0) no difference between first-wave ranked preferences and for whom a respondent voted. The third row, "Rank (Trump > Clinton)" is coded as: (+1) a shift from ranking Clinton preferable to Trump in the first survey wave to preferring Trump to Clinton in the second wave; (−1) ranking Trump preferable to Clinton in the first wave to preferring Clinton to Trump in the second wave; or (0) no change between survey waves. The left panel represents a coding of the variable of interest as the log exposures + 1 number of posts in a respondent's Twitter feed; the right panel, a binary variable indicating whether at least one such post appeared in a respondent's feed. $n = 1,496$ survey respondents.

## The relationship between exposure to Russian foreign influence accounts and voting behavior

We lastly turn to a key question raised by researchers, journalists, and politicians regarding the relationship between exposure to Russian foreign influence accounts and vote choice in the 2016 US presidential election. As with issue positions, we use within-subject variation in our outcomes by comparing voting preferences prior to the election campaign to voting preferences of those same respondents immediately prior to election day, and to vote choice in the election itself.

The first wave of the survey took place in April 2016, before the Democratic and Republican presidential nominees were decided. As a wave 1 measure of voting preferences, we therefore use respondents' rankings of the viable presidential candidates at the time to capture whether a respondent preferred Clinton to Trump, or vice versa. We then measure how respondents ranked Clinton and Trump in the survey wave immediately prior to the election, and how they voted in the election itself—a measure collected independently by YouGov following the election (see Supplementary Methods A2 for details of the survey and fielding schedule). From these data, we construct three related outcomes. First, we measure changes in voting preferences by comparing respondents' preference between Clinton and Trump prior to the election campaign to how those respondents voted in the actual election. Second, we measure changes in respondents' voting preference prior to the campaign to an equivalent survey-based measure taken in the last wave of the survey (immediately prior to the election). Third, we capture the possible broader relationship between exposure to posts from Russian foreign influence accounts and voting behavior by examining whether exposure benefited Trump more generally. To do so, we compare preferences for Trump over Clinton or vice versa in the first wave of the survey to whether a respondent switched from their first-wave preference by either voting for the other candidate; voting for a 3rd-party candidate; or abstaining from voting altogether. We use this third measure to assess the possibility that exposure to Russian foreign influence accounts is associated with disaffection from the Democratic nominee in ways that could have benefited the Trump campaign more generally. In sum, these measures capture vote choice in the election itself; survey-based candidate preferences; and a broader measure of voting behavior that measures benefits to one or the other candidate.

Using these measures, we investigate whether exposure to posts from Russian foreign influence accounts is associated with changes in preferences across survey waves (see caption of Fig. 5 and

Supplementary Methods E1 for more details on how variables are coded). For each outcome, we code a preference for Trump as 1, and a preference for Clinton as 0. A positive relationship would thus suggest that exposure to posts from Russian accounts is associated with a change in voting preferences or behavior favorable to Trump; a negative relationship, a change in preferences or behavior favorable to Clinton.

To examine the relationship between exposure to Russian foreign influence accounts and vote choice, we fit linear probability models that predict within-respondent changes in vote choice between the first and final wave of the survey conditional on exposure to posts from Russian foreign influence accounts. We fit two models per outcome, measuring exposure as the (log exposures + 1) number of posts from Russian foreign influence accounts and as a binary variable capturing whether a respondent was exposed to at least one such post during the campaign. We also fit an alternative model in which the final wave outcome is predicted conditional on first wave voting preference and exposure to posts from Russian accounts, with substantively equivalent results (Supplementary Methods E7).

Figure 5 presents the results. As estimates in the first panel indicate, the relationship between the number of posts from Russian foreign influence accounts that users are exposed to and voting for Donald Trump is near zero (and not statistically significant). This is the case whether the outcome is measured as vote choice in the election itself; the ranking of Clinton and Trump on equivalent survey questions across survey waves; and with the broader measure capturing whether voting behavior more generally favored Trump or Clinton through voting abstentions, changes in vote choice, or voting for a third party. The signs on the coefficients in each case are also negative, both for the count and binary measure, a result that would be inconsistent with a relationship of exposure being favorable to Trump. It is also worth noting that none of the other explanatory variables (with the exception of sex in some models) used as controls appear to be statistically significant predictors of the change in voting preferences (see Supplementary Methods E1, E2, E3).

To place the magnitude of the relationship between exposure to Russian accounts and voting preferences on a more interpretable scale, we simulate changes in voting preferences as a function of exposure. To do so, we simulate coefficients from each of the three models presented in the first panel of Fig. 5, and calculate the average difference between respondents' predicted change in voting preferences under their observed exposure to Russian foreign influence

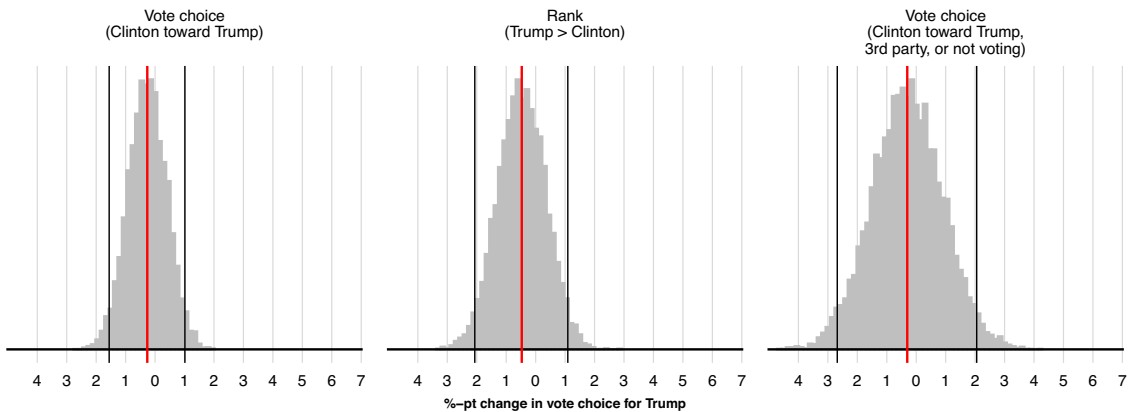

**Fig. 6 | Simulated change in vote choice under observed exposure to posts from Russian accounts relative to no exposure.** Simulated change in vote choice for each of the three vote preference outcomes. Vertical red lines indicates the median prediction. Vertical black lines indicate 90% prediction intervals.

accounts and that under the counterfactual of no exposure among any respondents. Negative values indicate changes in voting preferences in favor of Clinton; positive values, changes in favor of Trump.

Results are presented in Fig. 6. The red line in each panel indicates the median predicted change in voting preferences. The black lines indicate 90% prediction intervals. As the figure shows, the median predicted change in voting preferences when comparing the predicted change in vote for Trump under observed exposure relative to no exposure is near zero for each outcome. For vote choice in the election itself (first panel), for instance, the predicted change is a −0.18 percentage point decrease in vote for Trump (90% CI: −1.15, 0.78), and −0.4 for both the ranking based and more general vote choice measure. These estimates are broadly comparable to the ones reported in a large body of literature documenting minimal relationships from both offline and online campaigns[17] with reported results of a 0.7 percentage point change in voting choice from exposure to Facebook ads[31] or watching campaign videos[20]. Estimates from these distinctly targeted advertisements might be thought of as an upper bound on any theoretically achievable relationship from exposure to the Russian foreign influence campaign. Put differently, the minimal and non-significant relationships that we observe between exposure to posts from foreign influence accounts and voting behavior are similar in magnitude to the minimal and absent relationships observed in experimental research on the effects of traditional offline and online campaigns[18].

Given that the data are observational, we stress that the relationships that we estimate cannot be confidently said to be causal. However, as a descriptive exercise, we compare the observed relationships between exposure to Russian foreign influence accounts and the election results. In panel a of Fig. 6 the results show that the predicted change in vote for Trump was <0.7% points in 95% of the simulations when comparing observed exposure to the counterfactual of no exposure to Russian foreign influence accounts. By comparison, the closest margin of victory for Trump, in Wisconsin, was 0.77 percentage points, with Clinton also having needed to win multiple states with larger vote margins. We note that the intervals presented in Fig. 6 also represent 95% two one-sided test intervals[27,32], such that using Two One-Sided Tests for equivalence testing will reject relationships between exposure to foreign influence accounts and changes in vote choice that are greater than these bounds. Substantively, we note further that such minimal relationships are also consistent with causal evidence from a meta-analysis on the effects of targeted campaign advertising on vote choice[20], in which the authors find a null effect with a point estimate of 0.7 percentage points. Triangulating our results, the absence of meaningful relationships between exposure to Russian foreign influence accounts and voting behavior is also consistent with the evidence presented earlier that exposure to these foreign influence

accounts was concentrated, and was confined to a small group of respondents who identified themselves with the Republican Party and thus those least likely to vote for Clinton irrespective of exposure to the Russian foreign influence campaign.

## Discussion

There is widespread concern among researchers, journalists, and politicians that social media is being used by foreign governments to undermine elections in democracies worldwide. However, data limitations have precluded empirical investigation into the scale, prevalence, and effectiveness of these foreign influence campaigns on social media over the course of major election periods. In this study, we presented a systematic evaluation of foreign influence accounts during the 2016 US election. By linking a longitudinal survey administered over the course of the 2016 US election campaign to respondents' Twitter timelines, we provided an analysis of the relative scale of exposure to foreign influence campaigns as well as the potential relationships between the largest such campaign and users' issue positions, polarization, and voting behavior.

Taking our analyses together, it would appear unlikely that the Russian foreign influence campaign on Twitter could have had much more than a relatively minor influence on individual-level attitudes and voting behavior for four related reasons. First, we find that exposure to posts from Russian foreign influence accounts was concentrated among a small group of users, with only 1% of users accounting for 70% of all exposures. Second, exposure to Russian foreign influence tweets was overshadowed by the amount of exposure to traditional news media and US political candidates. Third, respondents with the highest levels of exposure to posts from Russian foreign influence accounts were those arguably least likely to need influencing: those who identified themselves as highly partisan Republicans, who were already likely favorable to Donald Trump. Fourth, we did not detect any meaningful relationships between exposure to posts from Russian foreign influence accounts and changes in respondents' attitudes on the issues, political polarization, or voting behavior. Each of these findings is not independently dispositive. Jointly, however, we find concordant evidence between exposure to Russian disinformation—which is both lower and more concentrated than one might expect to be impactful—and the absence of a relationship to changes in attitudes and voting behavior.

Nevertheless, despite these consistent findings, it would be a mistake to conclude that simply because the Russian foreign influence campaign on Twitter was not meaningfully related to individual-level attitudes that other aspects of the campaign did not have any impact on the election, or on faith in American electoral integrity. Importantly, the scope of our research is limited to the Russian foreign influence

campaign on Twitter. We also restrict our analysis to social media posts and thus cannot examine relationships from any potential sharing of other media content (e.g., images and videos) more generally. This research thus does not speak to the impact of similar campaigns on other social media platforms, nor to the possibility of foreign election interference via other channels, such as hacking or phishing schemes that were allegedly designed to surface information unfavorable to political opponents at opportune moments[10].

Finally, while our evidence points to the absence of a relationship between exposure to social media posts from Russian foreign influence accounts and individual-level outcomes, foreign influence campaigns may also succeed through second-order effects: those effects that are achieved by provoking a domestic reaction to the intervention itself[33]. Indeed, debate about the 2016 US election continues to raise questions about the legitimacy of the Trump presidency and to engender mistrust in the electoral system, which in turn may be related to American's willingness to accept claims of voter fraud in the 2020 election. Such beliefs appear to stem in large part from speculation that Russian interference—whether on social media or through other channels—influenced the election outcome[34–36]. In a word, Russia's foreign influence campaign on social media may have had its largest effects by convincing Americans that its campaign was successful[37]. Our results thus provide a corrective to the view that the foreign influence campaign and those like it can easily manipulate the attitudes and voting behavior of ordinary social media users. Foreign actors may nevertheless adapt their behavior on social media to have meaningful effects, and political contexts may become more conducive to foreign influence campaigns. This warrants that our results be taken with caution when assessing future foreign influence campaigns on social media.

## Methods

Our empirical investigation of exposure to social media posts from foreign influence campaigns is motivated by recent research on these campaigns[12], and relies on a three-wave longitudinal survey of US respondents that was conducted by YouGov, a major public opinion research firm. The survey data contain responses from 1496 US respondents who consented both to provide their Twitter account information for research purposes and to answer questions concerning their political attitudes and beliefs at multiple points during the 2016 US election campaign (Twitter-linked survey data collection was approved by NYU IRB 12-9058). The first wave of the survey was sent in April 2016; the last wave, immediately prior to the election at the end of October, 2016. Respondents were also recontacted after the election and asked to indicate whether they voted and, if so, for whom. The composition of survey respondents is approximately representative of the demographic profile of the US voting-age public (see Supplementary Table A2 in which we also compare our sample characteristics to estimates of the Twitter population published by the Pew Research Center). The survey instrument was designed to capture standard socio-demographic characteristics; attitudes toward a variety of election issues; and preferences over the political candidates in multiple survey waves. To capture the tweets that would appear in respondents' Twitter timelines during the election campaign, we collected the list of users whom respondents followed on Twitter, and retrieved all posts from these users that were sent in the eight months prior to election day (see Supplementary Methods A4 for further details). In aggregate, these tweets constitute 1.2 billion social media posts across all respondents. Finally, we use data released by Twitter to identify the posts in survey respondents' timelines that originated from the foreign influence campaigns that were active in the lead-up to the 2016 US election (see Supplementary Methods A1 for more details on Twitter data releases). Following similar studies[21], we refer to tweets that survey respondents were potentially exposed to in their timelines as their

"exposures". This is a limitation to the extent that although we can observe potential exposures (that is, tweets and retweets by accounts that a given users follows), we cannot know which tweets in their timelines users actually saw. However, examining potential exposures is currently the best practicable means to study the Russian foreign influence campaign—the most high-profile such campaign in recent history.

We also note a few other potential limitations of our data. First, because we rely on Twitter's identification of foreign influence campaign accounts, we cannot independently validate the identification process, which has not been detailed publicly by Twitter. However, the company has actively sought to identity, remove, and make public the accounts associated with foreign influence campaigns, which have since been successfully used in research that examines the behavior of accounts associated with these campaigns[7,8,13]. Second, although the survey panel data allow us to examine changes in political attitudes and preferences over time in response to exposure to Internet Research Agency tweets, the data do not approximate those from an ideal-case randomized experiment. Instead, given the obvious temporal, ethical, and legal constraints of randomly assigning posts from a foreign influence campaign to US social media users during an election campaign, the data represent a near best-case observational design. However, because the data are observational (not experimental), whether and how much a user is exposed to posts from Russian foreign influence accounts is not random. Foreign actors can be presumed, for example, to know their target audience and thus aim to maximize their influence by directing information toward certain users. The content of posts shared by Russia-sponsored accounts may also affect who is exposed to the extent that respondents may follow certain types of users who are amenable to the type of content shared by foreign influence campaigns. For example, recent studies have examined whether users interacted with Internet Research Agency accounts themselves[12,14], and foreign influence accounts have been shown to post information regarding specific political concerns (such as electoral fraud, or anti-establishment or racial issues[7,13]) that might drive some to be more exposed than others. Two aspects of the data help minimize these problems. First, because the survey data are panel data, we can model the relationship between exposure to posts from Internet Research Agency accounts and our outcomes of interest by using within-respondent variation in political attitudes, polarization, and vote preferences and behavior. We can thus account for time-invariant characteristics of survey respondents that would result in different levels of exposure. Second, concerns about selection would be strongest for social media users who choose to follow foreign state-sponsored accounts on Twitter. As we discuss in our results, most exposures to social media posts from Internet Research Agency accounts was indirect, either through retweets or quote tweets from other ordinary users. Exposure, in other words, was primarily incidental. This does not, however, mean that exposure is as-if randomly assigned to users. Even if users are exposed primarily incidentally, the choice to follow accounts that share particular types of content will not itself be made at random. In our regression analyses, we nevertheless test the robustness of our results to measuring exposure in three related ways: by the number of posts users were exposed to during the 2016 election campaign; by whether users were exposed to at least one post from a Russian Internet Research Agency account; and by whether a user followed at least one such foreign influence account (the last of which we detail in the Supplementary Methods E4). Our results are effectively equivalent for each of these three measures of exposure. In conclusion, we discuss the importance of triangulating the descriptive and regression-based results to arrive at a more complete picture of the role of the Russian influence campaign during the 2016 US presidential election.

## Software

To conduct the empirical analysis, we used R[38] and Python[39] programming languages, and the following R and Python libraries: *cowplot*[40], *data.table*[41], *dplyr*[42], *estimatr*[43], *fasttime*[44], *ggplot2*[45], *haven*[46], *kableExtra*[47], *knitr*[48], *lmtest*[49], *lubridate*[50], *magrittr*[51], *MASS*[52], *mvtnorm*[53], *numpy*[54], *pandas*[55,56], *readr*[57], *sandwich*[58,59], *stargazer*[60], *stringi*[61], *stringr*[62], *tibble*[63], and *tidyr*[64].

## Reporting summary

Further information on research design is available in the Nature Portfolio Reporting Summary linked to this article.

## Data availability

The authors have deposited computer code and minimal datasets (survey responses and aggregated exposure data) required to replicate the methods used in this paper in a GitHub repository located at: https://github.com/tpaskhalis/ncomms_russia_us_2016[65]. All other relevant data are available upon reasonable request to the authors. Full data are not publicly available due to terms of data license agreement signed with Twitter, Inc. and survey data containing information that could compromise research participant privacy. Source data are provided with this paper.

## Code availability

The code supporting this study is available at the following public GitHub repository: https://github.com/tpaskhalis/ncomms_russia_us_2016[65].

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

## Acknowledgements
We thank members of the Aarhus Research on Online Political Hostility project for their comments and suggestions. The Center for Social Media and Politics at New York University is generously supported by funding from the National Science Foundation, the John S. and James L. Knight Foundation, the Charles Koch Foundation, the Bill and Melinda Gates Foundation, the Hewlett Foundation, Craig Newmark Philanthropies, the Siegel Family Endowment, and NYU's Office of the Provost. This work was supported in part through the NYU IT High Performance Computing resources, services, and staff expertise.

## Author contributions
G.E., T.P., and J.T. designed the research, J.N., J.T., and R.B. designed the survey, G.E. and T.P. prepared the data for analysis, T.P. accessed the data on foreign influence accounts from the Election Integrity collection provided by Twitter, G.E., T.P., and J.Z. analyzed the data, G.E. and T.P. wrote the first draft of the paper, and all of the authors contributed to revising the manuscript.

## Competing interests
The authors declare no competing interests.
