## [Peer Review File · Nature Communications]

Exposure to the Russian Internet Research Agency foreign influence campaign on Twitter in the 2016 US Election and its relationship to attitudes and voting behaviorREVIEWER COMMENTS

Reviewer #1 (Remarks to the Author):

Referee report for "The Limited Impact of Russia's Election Interference on Twitter in the 2016 US Election" (Nature Communications 302261_0)

This study combines responses from a 1,500 person panel survey with information on exposure to Russian disinformation on Twitter. The study answers an important descriptive question well: how many and what kinds of people were exposed to Russian disinformation? The answer appears to be, the vast majority of people weren't exposed in a meaningful way and those that were tended to be strongly Republican. The study also examines a causal question: what is the average effect of exposure on political attitudes and vote choice. Here the answer appears to "approximately zero." I appreciated that the authors employed equivalence testing affirm small effects smaller than 0.2 SDs.

This paper is a needed correction to the too-commonly-held belief that Russian trolls stole the election for Donald Trump and I am largely supportive of publication.

My main concern is with the portion of the paper dedicated to causal inference. I'm not at all sure what the estimand here is -- the regression estimates seem to be shooting at an average treatment effect by comparing overtime changes among those with a lot of exposure to those with very little exposure. But the descriptive portion of the paper really points out that treatment was confined to a small group of people. I think the appropriate estimand therefore is an ATT among this group. In any case, I would urge the authors to be more specific about their causal estimand.

I also think that the paper does only an OK job describing the identification approach. Yes, we are calculating "within-unit" changes, but the inference comes by comparing treated to untreated. The pre-treatment value of the outcome variable is just a covariate here like any other. We could just as easily include it as a right-hand side regressor, rather than explicitly differencing (differencing sets the coefficient on this variable to 1, whereas regression will choose the best fitting value).

Small notes

- tiny style comment: many sentences and clauses begin with "there is" or "there are".
- On page 7, the manuscript suggests that selection may not be a concern because exposure was indirect. I don't like this line of reasoning -- even if subjects don't "self-select," the selection forces that generate indirect exposure may confound our causal inferences just as strongly!

Reviewer #2 (Remarks to the Author):

Review of "The Limited Impact of Russia's Election Interference on Twitter in the 2016 US Election"

This paper examines the scope and effects of exposure to Twitter posts originating with accounts linked to Internet Research Agency (IRA) accounts during the 2016 election. The approach is to use a longitudinal survey of ~1,500 Americans that then is linked to their Twitter accounts, allowing analysis of what content they might have seen over the course of the election.

The use of the nationally representative, longitudinal survey sample is an appealing feature of this work. Even for the descriptive aspects of this work that aim to document the degree of potential exposure (my favorite part of the paper), this can have important advantages over approaches that would lack a way to confirm whether sampled Twitter accounts belong to individual Americans (or, if one does so, as in ref. 10, then this may create an importantly biased sample).

Exposure

I found the analyses of potential exposure generally interesting and worthwhile. The figures were generally well designed displays of these views into how many people may have seen these tweets via their follower networks and in what quantities. The comparison with other foreign-government-backed campaigns was helpful. For example, I generally quite liked Figure 2 (though I would note that daily medians may be less informative since as the bin size — days — gets smaller, any differences eventually disappear).

One interesting point is about the concentration of exposures. As the authors note, this is similar to ref. 10. This made me wonder, is this just true of many content types/sources on Twitter? What is the concentration for exposures to content about particular celebrities? Or even just politicians in Figure 2A?

The section on what predicts exposure is interesting in how largely null the results were except for partisanship, gender, and social media use. Is there more about baseline behaviors and exposures on Twitter that is predictive here? For example, were there “big” accounts that people follow or domains that people shared links from that predicted high eventual exposure? This is interesting in its own right, but would also be important for arguing about how confounded “incidental” exposures are.

I worried about how much exposure, even on Twitter, this analysis might be missing. My understanding (e.g., from the Senate-commissioned reports) is that there was a lot of image-based “meme” content produced and/or disseminated by IRA-linked accounts. While retweets are tracked here, it seems like if an image gets downloaded and uploaded that provenance is not tracked. One might worry that a substantial part of the exposure is to memes that turned out to be particularly popular and spread widely without explicit retweet links back to accounts identified as IRA-linked. This would, at least, attenuate any effects. Maybe there is some way to estimate or bound how much of the total exposure these kinds of exposures are.

Effects of exposure

An important aim of this paper is to say something about the effects of exposure to IRA campaigns on political attitudes and participation. The conclusion is that any such direct effects are “limited” and that they likely had no more than “a very minor effect on individual-level attitudes and voting behavior”. I can’t say I updated my beliefs much based on this, for reasons I describe below.

What effects are ruled out?

I found the comparison of the effect sizes included in the CIs to old, generic effect size categories for Cohen’s d unconvincing, even absurd in the present context since I think they would suggest nearly all campaign activities (even door-to-door contact) may be “negligible”. When it comes to voter turnout and vote choice, we have a lot more to go on to benchmark effects. If anything, the analysis of effect after transformation to standardized effect sizes (in SI section B) obscures rather than illuminates. To be explicit, the paper says (on one definition) that effects below 36% of a standard deviation be regarded as negligible. It wasn’t immediately clear to me what these are standard deviations of. Consider the vote choice outcome; is this normalized by the standard deviation of vote choice or change in vote choice? (If the latter, this is a reason not to like standardized effect sizes, since it would change the categorization of the effect size based on whether one collected baseline outcome data or not, even when the research question remains unchanged. I don’t think this standard deviation is reported anywhere, but maybe I missed it.)

I said we have more to go on when thinking about effect sizes in voter turnout and vote choice. (I also focus on these outcomes because I am likewise more familiar with them.) First, we have other estimated effects to compare with, whether from impersonal contact (where they are typically but not always small) or from other interventions. If the results are compatible with effects much, much larger

than those from, say, TV ads, then it isn't clear how much has really been ruled out. Some of this literature is reviewed in the introduction, but not very quantitatively, and it is not returned to in order to contextualize these results. If I understand the results correctly (Table D12), the 95% confidence intervals for the effect of at least one exposure on vote choice are approximately $4 \times 0.03 = 0.12$ wide; that is, the range of effects they are compatible with is 12 percentage points wide! Given our priors about direction, notably this doesn't rule out a >3pp effect in Trump's direction. Similar calculations could be done for the number of exposures.

Second, we have benchmarks to compare with for possible effects on the outcome of elections: How large of an effect would one need to tip a contest? Note that altering the outcome of, e.g., congressional contests could require fewer voters than the presidential race. It would also seem that evidence about geographic concentration of exposure would be important here. (And maybe this is a thing to note from the prior part: Exposure seems more geographically diffuse. Perhaps it is useful to consider a swing-state vs. not categorization, rather than a traditional regional one.)

The authors should certainly know that plausible effects here are small. I think if you asked political scientists who have worked on advertising effects to guess at the effect sizes of these kinds of exposures they would all — including those who think such exposures could plausibly tip election outcomes! — guess effects that are smaller than 36% of a standard deviation. So hard to know what we should take away from this, even if we believe that these are unbiased estimates of causal effects of exposure.

Observational causal inference

The aim of this part of this paper is inference about the effects of exposure to IRA-linked accounts. The identification strategy is based on the longitudinal data. (However, it doesn't strictly account for any time-invariant confounding since the estimation approach is to use change scores as outcomes, rather than, say, including respondent fixed effects.)

There is an (entirely informal, I think) argument that much of the exposure is incidental via retweets, rather than following the accounts and that there shouldn't be confounding in this. I think this is an interesting idea here, and one that perhaps could be fleshed out more. First, this argument could be made quantitative. For example this might include (a) characterizing in more detail how much exposure occurs incidentally in this way (and who it is occurring to, going beyond Figure C10, so as to argue this is as good as random) and (b) treating the results in Appendix D as informative about the amount of confounding. Second, something like this argument would justify the idea of using characteristics of friends of friends as instrumental variables (Bramoullé et al. 2009 and subsequent literature); however, it is commonly understood there that this makes the strong assumption that friends-of-friends' characteristics are exogenous. That is, perhaps whether, say, someone follows someone who follows an IRA-linked account is less confounded than whether someone follows an IRA-linked account directly, but there certainly could still be plenty of selection here.

I guess maybe I think this isn't worth digging into more because the whole thing is underpowered, but I could imagine trying to improve all of this. For example, by including placebo tests for effects of future exposures, or arguing that certain sources of exposures either should be adjusted away or are as good as random and so could be, at least theoretically, used as instruments.

In summary, I think this paper is strongest as a descriptive study of exposure to IRA-linked accounts. And I do very much think this is worthwhile and that the linked survey data is helpful here.

I'm not sure the reader learns much about effects of exposure, both because of confounding and because the sample sizes make the results too imprecise to be informative. I think even with other improvements it is unclear to me that it can be made more informative. This problem is not unique to

this work. Leading political scientists have sometimes been confused by underpowered studies of digital campaign activities (Broockman & Green, 2014; cited in a very relevant context in ref. 1 where there is a lot of attention paid to the challenges of being sufficiently powered to study this). And other studies of exposure to IRA-linked accounts [2] have similarly been uninformative because they were dramatically underpowered. So maybe the best use of the latter half of the paper is to rework it as a warning to others that even an impressive panel like this is often going to be largely uninformative. Maybe the authors think differently about this. That's fine, but it seems like then the paper would really need to incorporate more detailed arguments about why the results are informative and rule out plausible effect sizes and/or effect sizes that would have electoral consequences.

Overall, I like this effort and think that the first part is valuable, but the second part is just doesn't really advance our knowledge — and may confuse people — as is.

Sincerely,
Dean Eckles

Other comments:

- I think a lot of the references to the SI/appendices are off throughout. Like p. 9 says to look to Figure B8 for the share of exposures that are indirect, but I think this is actually Figure C10.
- I think it would be better to use a quasi-Poisson model than to do the log-transformation of the number of exposures when used as an outcome (what is happening with the 30% of people with 0s?).
- Similarly, when the number of exposures is a predictor, perhaps it could make sense to estimate a model with both an indicator for any exposure and $\min(0, \log(\text{exposures}))$ rather than a single term for $\log(\text{exposure}+1)$, which I assume is what used, though everything just says ' $\ln(\text{exposures})$ '.
- What's the reason for focusing on change scores (pre-post differences) rather than using regression adjustment? The former is a special case of the latter with the slope(s) fixed to 1, which is rarely the optimal choice. (Incidentally, this also creates the confusion I experienced about the standardized effect sizes, though I would also suggest just removing or substantially deemphasizing the standardized effect sizes.) As noted above, a fixed-effects approach could also make sense, as could other methods to more flexibly control for baseline behaviors and interests (Eckles & Bakshy 2020), though perhaps the sample sizes are again too small here.
- "As Panel A of Figure 1 shows further, posts from the Russian foreign influence campaign were by far the most prevalent, representing 86% of all exposures in respondents' timelines in the lead-up to the election." (p. 8) Could be made more clear by noting this is 86% of the foreign influence campaigns included here.
- Are self-reports of turnout reliable enough for the purposes they are put to here?

References

- Bramoullé, Y., Djebbari, H., & Fortin, B. (2009). Identification of peer effects through social networks. *Journal of Econometrics*, 150(1), 41-55.
- Eckles, D., & Bakshy, E. (2020). Bias and high-dimensional adjustment in observational studies of peer effects. *Journal of the American Statistical Association*.

Reviewer #3 (Remarks to the Author):

The manuscript presents a well-designed study on an important topic. The wide-ranging speculations about the impact of Russian interference on the 2016 elections have so far been backed by little direct evidence. The authors make a good case for the minimal effects of the interference campaign, at least as far as changing the minds of Twitter users is concerned.

I only have a couple of more significant concerns to mention about the methods and conclusions of the paper. The first one of these has to do with the main dependent variable, exposure to IRA

misinformation. While the study consistently speaks of “exposure”, the actual measure used should more accurately be named “potential exposure” or “network exposure”. This may be good to explain early in the study as it does not become fully clear to the reader until the methods section. The measure used here is based on the number of IRA-sourced tweets posted by accounts that the respondent follows. There are a couple of caveats to this metric: (1) A person is not guaranteed to have seen all tweets from their social network (or even to have had those tweets included in their stream). This would suggest many of the IRA tweets may not have been seen by the users; and (2) Twitter users often follow information beyond their immediate networks, e.g. by following political hashtags. This would suggest that users who are considered “not exposed” could actually have seen IRA tweets.

This is all not to say that the metric used by the authors is not meaningful – it is more of a caution that it might be a noisy measure, even when aggregated over multiple months. I would suggest that the authors explicitly discuss this as a limitation of the study, as it could plausibly be one cause for finding null effects from misinformation exposure.

Relatedly, I would suggest one small tweak in the regression models presented in the paper. That would be to include as a control not only “social media use”, but also the overall volume of tweets from the user’s network. The reasoning is that a large volume of tweets from followed accounts reduces the probability of seeing any individual tweet. I also did not see a description of how “social media use” is measured or how it was distributed among respondents – I may have missed it but if not, that would be good to include.

Somewhat surprisingly, the study never discusses user interactions with misinformation: e.g. the number of times people in the sample favorited, retweeted, or posted IRA misinformation materials. Based on previous research, it is likely that only a small percent of the respondents ever did that. It would nonetheless be useful to report it. As these behaviors send a very strong signal of interaction with misinformation, it would also be good to examine within-subject changes for those respondents in particular.

A couple of questions I have about the study revolve around the regression models examining potential effects from IRA misinformation exposure.

First, previous research has shown that political misinformation in 2016-2020 has heterogeneous effects on consumer attitudes depending on ideology. Exposure to the same misinformation item can shift the attitudes of strong Republicans in one direction, and that of strong Democrats in the opposite direction. Thus, it is plausible that the effect for the full sample could appear null, while there are significant but heterogeneous effects of exposure within groups. I would suggest looking at those groups separately to rule this out.

In addition, as the authors point out, the misinformation content itself may have served a combination of goals. The agency is said to have spread multiple versions of stories with diverging messages meant for different audiences. Not sure if there is any way plausible way to identify/label the type of content users were exposed to (perhaps identifying tweets that contain prominent hashtags representing different political views, etc). If that is possible, it would be quite valuable and interesting.

Also important, I would recommend reconsidering the race/ethnicity treatment in the study. The authors examine race using the “White” vs. “Non-White” categories. It would be better to disaggregate Non-Whites and look at specific groups. This is especially key given that Russian misinformation seems to have exploited racial sentiments in the US. As can also be seen in the hashtag word cloud included in the supplementary materials, the IRA is known to have produced a large number of messages related to the Black Lives Matter movement. With that in mind, looking at Black respondents separately seems indicated.

One final note about the comparison of IRA troll tweets with other types of content. I found that interesting and informative, but my takeaway differs somewhat from that of the authors. The study

notes that tweets from U.S. politicians outweigh those from Russian trolls 9:1. To me, one troll tweet for every nine politician tweets seems quite high, especially given that we are talking about only one of the bad actors out there vs. the entire American political elite. One thing that would be an interesting point of comparison to offer the readers is a chart showing the concentration of exposure to U.S. politicians (similar to Fig.1 B and C). That could help us better understand how the direct reach of legitimate political actors may compare to that of foreign misinformation sources.

Reviewer #4 (Remarks to the Author):

This is an interesting and important paper which provides an analysis of the characteristics of U.S. twitter users who may have seen posts from the Russian interference efforts in the 2016 election. Broadly speaking the authors find that (1) the vast majority of potential exposures were to a small minority of accounts, (2) strong Republicans were the ones most likely to be following accounts that posted or reposted the material, and (3) that accounts which conceivably would have seen more exposure to posts were not owned by people experiencing substantial opinion shifts between April and October 2016.

I quite like this paper and would want to see it published. It is at its best when it is presenting descriptive findings and providing interpretation. My suggestion is primarily on how to maintain that focus on description and avoid overclaiming evidence for causal effects. I split comments into a few thematic sections below.

(1) Descriptive Evidence and Claims

The paper makes a series of clearly descriptive claims (e.g. Figure 3, Panel A depicting disproportionate exposure of posts to Republicans) and a series of claims about 'effects' (Figure 3, Panel B showing a regression of exposure on party ID and 'controls'). These latter elements are not as well motivated as the descriptive claim and sit in an uncomfortable space between causal inference and description. For example, the panel B regression is motivated with the statement "To examine whether the findings in Panel A are an artifact of factors other than partisanship, we fit an OLS regression model to predict the (log) number of posts from Russian trolls that were in the timelines of each US respondent during the 2016 election campaign." Here I think the authors should make the claim and the relevance of this conditional expectation more explicit. I take the claim to be that they believe, somewhere, in some room, there was a conversation where there was a decision to target Republicans. They (implicitly) argue that an observable implication of that would be not just that the marginal distribution was more strongly tilted republican, but also that across demographic subgroups, republicans consistently saw more exposure. Having a small sample, they approximate these subgroups with a regression and then average over them. I don't think you have to put all of that on the page, but I do think if this is the train of thought more of it should be represented. As it is now, it seems like a 'regression that controls for things' because that is what is expected.

To the same end I think more humility is necessary in the claims around effects of exposure. Again, the authors implicit argument would seem to be that if exposure to this material caused a shift in opinions we would see that people with higher exposures did not move more in their opinions than people with low exposures. (Note: I think this is what is going on but if I am correct there is an error in the caption of figure 4 which describes each as a model of exposure on ideology rather than the other way around. This is not helped by the fact that a similar style of plot early has independent variables down the rows rather than outcomes). This is a helpful descriptive implication of the authors purported state of the world, but it should be clearly stated as such rather than presented implicitly as an estimate of a causal effect.

This is all to say that the authors are making claims in a difficult situation where good data is scarce,

but that's all the more reason to make the claims humbly and with a clear presentation of the inferential jumps that are required to believe the case. To the extent possible I would also be careful about how these findings are presented in the abstract as we know in practice many people won't get further than that.

(2) Measurement Error and Null Findings

The authors are in the unenviable position of arguing for null findings in a setting with a truly astonishing amount of measurement error. Ostensibly the construct is not whether a post from an account appeared in a timeline, but whether the individual actually read or engaged with the post. Because the measurement error of this construct isn't classical, it is hard to know in what way it biases results, but I think it is reasonable to suppose that because the measure is almost certainly an overestimate of real engagement with these particular posts, it is plausible that these obscure any real effect on a small subpopulation of users. In this sense it is very plausible that the measurement error leans in the direction of the author's finding which is an uncomfortable place to be. I'm not sure this is avoidable but it feels like it should be acknowledged.

I also applaud the authors for the equivalence testing but I think this section would be more readable and informative if the argument was made more substantively rather than using conventional fractions of standard deviations to think about negligible size.

(3) Bringing Caveats Forward

A number of very reasonable caveats and limitations to the work are mentioned in the final paragraph of the paper. I think this comes too late. The authors begin the final paragraph "Despite these findings, it would be a mistake to conclude that simply because Russian IRA trolls activity on Twitter did not meaningfully impact individual-level attitudes that other aspects of the Russian foreign influence campaign did not have any impact on the election or faith in American electoral integrity." But whether intended to or not, this is the opposite of the impression conveyed throughout the entire rest of the paper. Even this statement I think overstates the evidence available for identifying the 'meaningful impact' on attitudes. I think these points need to be raised earlier.

This work is important- even with the caveats stated clearly and prominently earlier in the paper. At least for my part, given the limitations of the argument more prominent would make the paper more credible, not weaker.

(4) Smaller Details

(a) when presenting the survey the paper says that "The composition of survey respondents is approximately representative of the demographic profile of the US voting-age public" and does not further raise the issue. This strikes me as not obviously ideal given that the inference is about the Twitter population and not the U.S. population which is notably different. Perhaps it would be possible to discuss explicitly what the target population is?

(b) The authors write: "Finally, we use data released by Twitter to identify the posts in survey respondents' timelines" Can you provide additional details on exactly what this procedure is and what it does and does not capture? It would be very helpful to clarify what we think this translates to in terms of rates of people actually reading posts. My guess is that the authors don't necessarily know, but being more expert than the readers on the subject, even providing some guidance of plausible values would be helpful.

(c) I think Figure 2 panel B has a mislabeled Y-axis and should be median?

(d) The reference to appendix B3 in the text should (I think) be C3 as there is no B3 and C3 would seem to be the correct material.

(e) Appendix G (the software statement) is a wonderful contribution.

Overall:

This paper is important, informative and likely to be hugely influential. While I think the authors make a compelling case with imperfect evidence, I also think they misrepresent the strength of that evidence at times particularly for a lay reader. Being clearer about the limitations (and why stronger evidence will likely never be available) seems like it would only make the paper stronger.

Reviewer 1 comments

This study combines responses from a 1,500 person panel survey with information on exposure to Russian disinformation on Twitter. The study answers an important descriptive question well: how many and what kinds of people were exposed to Russian disinformation? The answer appears to be, the vast majority of people weren't exposed in a meaningful way and those that were tended to be strongly Republican. The study also examines a causal question: what is the average effect of exposure on political attitudes and vote choice. Here the answer appears to "approximately zero." I appreciated that the authors employed equivalence testing affirm small effects smaller than 0.2 SDs.

This paper is a needed correction to the too-commonly-held belief that Russian trolls stole the election for Donald Trump and I am largely supportive of publication.

My main concern is with the portion of the paper dedicated to causal inference. I'm not at all sure what the estimand here is – the regression estimates seem to be shooting at an average treatment effect by comparing overtime changes among those with a lot of exposure to those with very little exposure. But the descriptive portion of the paper really points out that treatment was confined to a small group of people. I think the appropriate estimand therefore is an ATT among this group. In any case, I would urge the authors to be more specific about their causal estimand.

We thank the reviewer for requesting that we be clearer about what the regression models aim to estimate. Our overall goal in the manuscript is to triangulate evidence from (1) the high concentration of exposure, (2) the type of user who is exposed, and (3) the relationships between exposure and a number of political outcomes. In the revised manuscript we have sought to clarify the latter of these, for example on page 17 we note that the relationships that we observe are driven by those who are actually exposed to Russian influence accounts (which we agree cannot be interpreted as an ATE). As a related point, we also now make clear that one might observe different relationships if the people

who were exposed were, counterfactually, respondents who were Democrats or Independents. We highlight this because, as the reviewer suggests, a large part of our understanding of Russia's influence campaign that emerges from these data is that those who were exposed tended to be users who were already highly likely to support the Republican candidate.

I also think that the paper does only an OK job describing the identification approach. Yes, we are calculating “within-unit” changes, but the inference comes by comparing treated to untreated. The pre-treatment value of the

outcome variable is just a covariate here like any other. We could just as easily include it as a right-hand side regressor, rather than explicitly differencing (differencing sets the coefficient on this variable to 1, whereas regression will choose the best fitting value).

We thank the reviewer for highlighting this point regarding the use of change scores as compared to adding the pre-treatment measure as a right-hand side covariate. Our choice of using change scores was guided by the fact that in an observational setting like ours in which exposure is not randomly assigned—as the reviewer rightly highlights above concerning the ATT—we have natural concerns about confounding. Our choice to use change scores stemmed from evidence that change scores are shown to produce less biased estimates when the threat from confounding is larger (on this, we were originally guided by this simulation study (DeclareDesign, 2019) by the authors of Blair et al. (2019)). As a result of the reviewer's comment, we fit regression models for the issue positions, polarization, and vote choice outcomes using the Wave 1 measure of each outcome as a RHS variable, with substantively similar results. We have not included these in the Appendix because they have higher potential to be biased, but we are happy to include them if the reviewer suggests this would be useful.

Small notes

- tiny style comment: many sentences and clauses begin with “there is” or “there are”.

In the revised manuscript, we have sought to rephrase a number of these sentences to avoid this kind of repetition.

- On page 7, the manuscript suggests that selection may not be a concern because exposure was indirect. I don't like this line of reasoning – even if subjects don't “self-select,” the selection forces that generate indirect exposure may confound our causal inferences just as strongly!

We agree with the reviewer that the implicit claim in this section is too strong. In the revised manuscript we have made explicit the fact that users who are incidentally exposed to foreign influence content will also have self-selected into following users who—while not foreign influence accounts themselves—are a certain type of user who would share such content. We are also more cautious in our claims overall about interpretations of our regression results as causal, noting a number of limitations of the data regarding this throughout.

Reviewer 2 comments

This paper examines the scope and effects of exposure to Twitter posts originating with accounts linked to Internet Research Agency (IRA) accounts during the 2016 election. The approach is to use a longitudinal survey of 9,1,500 Americans that then is linked to their Twitter accounts, allowing analysis of what content they might have seen over the course of the election.

The use of the nationally representative, longitudinal survey sample is an appealing feature of this work. Even for the descriptive aspects of this work that aim to document the degree of potential exposure (my favorite part of the paper), this can have important advantages over approaches that would lack a way to confirm whether sampled Twitter accounts belong to individual Americans (or, if one does so, as in ref. 10, then this may create an importantly biased sample).

Exposure

I found the analyses of potential exposure generally interesting and worthwhile. The figures were generally well designed displays of these views into how many people may have seen these tweets via their follower networks and in what quantities. The comparison with other foreign-government-backed campaigns was helpful. For example, I generally quite liked Figure 2 (though I would note that daily medians may be less informative since as the bin size gets smaller, any differences eventually disappear).

We thank the reviewer for this comment, and we agree that the descriptives provide an especially compelling angle for understanding the role of the Russian foreign influence campaign during the election. To address the comment regarding fine-grained bin sizes, in our revision we have changed Panel B to show weekly rather than daily medians. Given that even these wider bins still result in a median of zero for exposure to Russian foreign influence accounts, we note the reason for this explicitly in the manuscript (that exposure is heavily concentrated among a small number of users) to ensure that this is understood by the reader. We could further widen the bin size. However, the fact that the median is zero across an even wider time window acts as a visually compelling feature of this figure by demonstrating the dramatic difference in exposure among ordinary US citizens to posts from Russian IRA accounts and traditional political actors. Both the relative amount of exposure (as presented in Figure 2) and its concentration (Panel B of Figure 1) provide a key piece of the foundation for the manuscript's argument concerning the Russian foreign influence campaign's potential overall scope and consequences.

One interesting point is about the concentration of exposures. As the authors note, this is similar to ref. 10. This made me wonder, is this just true of many

content types/sources on Twitter? What is the concentration for exposures to content about particular celebrities? Or even just politicians in Figure 2A?

This is an excellent suggestion, and led us make an important comparison in the concentration of exposure to Russian foreign influence accounts relative to exposure to news media organizations and politicians. In the revised manuscript we have added the requested graphs to SI Appendix D, which we reproduce as a figure below. In the revised manuscript we note that the concentration of exposure to foreign influence accounts is especially pronounced, and not equivalent to political communication on Twitter more generally: analogous concentration plots for politicians and news media show that 1% of respondents account for 24% and 37% of exposures to posts from news media organizations and politicians respectively. This compares with 1% of respondents accounting for 70% of exposures to Russian foreign influence accounts. Similarly, 10% of respondents accounts for 67% and 79% of exposures to posts from news media and politicians, compared to effectively all (98%) of exposure to IRA accounts. Exposure to political and news media accounts is indeed relatively high. But exposure to Russian foreign influence accounts is especially concentrated among an exceptionally small subset of users.

The section on what predicts exposure is interesting in how largely null the results were except for partisanship, gender, and social media use. Is there more about baseline behaviors and exposures on Twitter that is predictive here? For example, were there “big” accounts that people follow or domains that people shared links from that predicted high eventual exposure? This is interesting in

its own right, but would also be important for arguing about how confounded “incidental” exposures are.

We too see this as a potentially interesting analysis and we thank the reviewer for this suggestion. To examine this, we analyzed the predictive role of the followership network by fitting an L1 regularized logistic model with a binary variable indicating any type of exposure to Russian influence accounts as the outcome and accounts of news media and politicians (971 accounts in total) as predictors. The table below presents the top 10 Twitter accounts that were predicted to have the largest coefficients in this model. While certain accounts here confirm our expectations, with Republican politicians' accounts, including Donald Trump, being some of the more important features, overall this does not appear especially informative over and above the results in Figure 3: the model shows that following larger political news media and well-known politicians is predictive of exposure. We have thus not included these results in the Appendix. However, if the reviewer thinks this would be a worthwhile addition, we would of course add it in a further revision.

user_id	coefficient	user_screen_name
51241574	0.1912248	AP
428333	0.2390258	cnnbrk
7309052	0.2879663	yahoonews
14075928	0.5662732	TheOnion
1339835893	0.8653261	HillaryClinton
15745368	0.2597790	marcorubio
216881337	0.3432988	RandPaul
25073877	0.5098212	realDonaldTrump
10774652	0.4870633	theblaze
18510860	0.6520844	motherjones

I worried about how much exposure, even on Twitter, this analysis might be missing. My understanding (e.g., from the Senate-commissioned reports) is that there was a lot of image-based “meme” content produced and/or disseminated by IRA-linked accounts. While retweets are tracked here, it seems like if an image gets downloaded and uploaded that provenance is not tracked. One might worry that a substantial part of the exposure is to memes that turned out to be particularly popular and spread widely without explicit retweet links back to accounts identified as IRA-linked. This would, at least, attenuate any effects. Maybe there is some way to estimate or bound how much of the total exposure these kinds of exposures are.

We thank the reviewer for noting this possibility. We have to admit that we cannot collect data like these to identify content that was introduced to the eco-system by Russian foreign

influence accounts and then shared independently by accounts that were not IRA accounts. To clarify this for readers, we now note in our Discussion section that a limitation is that our analysis is restricted to social media posts from foreign influence accounts themselves and not content shared by way of other rich media content (e.g. images and video). It is worth noting that we would expect retweets and sharing of social media posts to be the dominant form of spreading content on Twitter because, as a platform, it is geared toward this type of engagement. We cannot, however, wholly rule out the possibility of other avenues of content sharing from external sources linked to the influence campaign.

Effects of exposure

An important aim of this paper is to say something about the effects of exposure to IRA campaigns on political attitudes and participation. The conclusion is that any such direct effects are “limited” and that they likely had no more than “a very minor effect on individual-level attitudes and voting behavior”. I can't say I updated my beliefs much based on this, for reasons I describe below.

What effects are ruled out?

I found the comparison of the effect sizes included in the CIs to old, generic effect size categories for Cohen's d unpersuasive, even absurd in the present context since I think they would suggest nearly all campaign activities (even door-to-door contact) may be “negligible”. When it comes to voter turnout and vote choice, we have a lot more to go on to benchmark effects. If anything, the analysis of effect after transformation to standardized effect sizes (in SI section B) obscures rather than illuminates. To be explicit, the paper says (on one definition) that effects below 36% of a standard deviation be regarded as negligible. It wasn't immediately clear to me what these are standard deviations of. Consider the vote choice outcome; is this normalized by the standard deviation of vote choice or change in vote choice? (If the latter, this is a reason not to like standardized effect sizes, since it would change the categorization of the effect size based on whether one collected baseline outcome data or not, even when the research question remains unchanged. I don't think this standard deviation is reported anywhere, but maybe I missed it.)

I said we have more to go on when thinking about effect sizes in voter turnout and vote choice. (I also focus on these outcomes because I am likewise more familiar with them.) First, we have other estimated effects to compare with, whether from impersonal contact (where they are typically but not always small) or from other interventions. If the results are compatible with effects much, much larger than those from, say, TV ads, then it isn't clear how much has really been ruled out. Some of this literature is reviewed in the introduction, but

not very quantitatively, and it is not returned to in order to contextualize these results. If I understand the results correctly (Table D12), the 95% confidence intervals for the effect of at least one exposure on vote choice are approximately $4 \times 0.03 = 0.12$ wide; that is, the range of effects they are compatible with is 12 percentage points wide! Given our priors about direction, notably this doesn't rule out a >3pp effect in Trump's direction. Similar calculations could be done for the number of exposures.

Second, we have benchmarks to compare with for possible effects on the outcome of elections: How large of an effect would one need to tip a contest? Note that altering the outcome of, e.g., congressional contests could require fewer voters than the presidential race. It would also seem that evidence about geographic concentration of exposure would be important here. (And maybe this is a thing to note from the prior part: Exposure seems more geographically diffuse. Perhaps it is useful to consider a swing-state vs. not categorization, rather than a traditional regional one.)

The authors should certainly know that plausible effects here are small. I think if you asked political scientists who have worked on advertising effects to guess at the effect sizes of these kinds of exposures they would all including those who think such exposures could plausibly tip election outcomes! guess effects that are smaller than 36% of a standard deviation. So hard to know what we should take away from this, even if we believe that these are unbiased estimates of causal effects of exposure.

We agree with the reviewer that the use of Cohen's d to quantify the relationship between exposure to posts from foreign influence accounts and outcomes like vote choice and turnout obscures the meaning of the estimates. This was done in the original manuscript because equivalence testing in political science has tended to benchmark in terms of standard deviations. In the revision, we have re-conducted the analysis to place our estimates on a substantively meaning scale (i.e. percentage point change), and used simulation to capture uncertainty in the estimated relationships in terms of percentage point change as a function of exposure. To do this, we predicted the mean change in voting by fitting the voting behavior models and then calculating the predicted change in vote choice first by setting exposure for all respondents in the dataset to 0, and then setting exposure to the actual amount of exposure respondents received in the observed data. This provides an estimate of the relationship between exposure and voting in the election under the counterfactual that no respondent was exposed to any content from Russian foreign influence accounts (compared to actual exposure, which we know to be heavily concentrated among Republicans). We show these results graphically in Figure 6 in the revised manuscript, and have reproduced them below, where the vertical

red lines indicate the median prediction, and vertical black lines indicate 90% prediction intervals. Importantly, as we also note further below, we are also now more cautious in how we discuss the regression results with respect to the limits of assigning a strong causal interpretation to the results.

As the reviewer notes, in the original draft of the manuscript we cite the literature regarding the size of campaign effects in the theoretical section, but do not return to them in the results section. Thus, rather than benchmark against standard deviations, we now follow the reviewer's suggestion to benchmark estimates to the recent empirical literature on campaign effects, and also compare them to the vote margins from the 2016 US presidential election results by way of example. In the revised manuscript, we have added comparisons to effect sizes reported in the political science literature (Kalla 2018, Broockman 2014, Coppock 2020).

Observational causal inference

The aim of this part of this paper is inference about the effects of exposure to IRA-linked accounts. The identification strategy is based on the longitudinal data. (However, it doesn't strictly account for any time-invariant confounding since the estimation approach is to use change scores as outcomes, rather than, say, including respondent fixed effects.)

There is an (entirely informal, I think) argument that much of the exposure is incidental via retweets, rather than following the accounts and that there shouldn't be confounding in this. I think this is an interesting idea here, and one that perhaps could be fleshed out more. First, this argument could be made quantitative. For example this might include (a) characterizing in more detail how much exposure occurs incidentally in this way (and who it is occurring to, going beyond Figure C10, so as to argue this is as good as random) and (b) treating the results in Appendix D as informative about the

amount of confounding. Second, something like this argument would justify the idea of using characteristics of friends of friends as instrumental variables (Bramoullé et al. 2009 and subsequent literature); however, it is commonly understood there that this makes the strong assumption that friends-of-friends' characteristics are exogenous. That is, perhaps whether, say, someone follows someone who follows an IRA-linked account is less confounded than whether someone follows an IRA-linked account directly, but there certainly could still be plenty of selection here.

I guess maybe I think this isn't worth digging into more because the whole thing is underpowered, but I could imagine trying to improve all of this. For example, by including placebo tests for effects of future exposures, or arguing that certain sources of exposures either should be adjusted away or are as good as random and so could be, at least theoretically, used as instruments.

In summary, I think this paper is strongest as a descriptive study of exposure to IRA-linked accounts. And I do very much think this is worthwhile and that the linked survey data is helpful here.

I'm not sure the reader learns much about effects of exposure, both because of confounding and because the sample sizes make the results too imprecise to be informative. I think even with other improvements it is unclear to me that it can be made more informative. This problem is not unique to this work. Leading political scientists have sometimes been confused by underpowered studies of digital campaign activities (Broockman & Green, 2014; cited in a very relevant context in ref. 1 where there is a lot of attention paid to the challenges of being sufficiently powered to study this). And other studies of exposure to IRA-linked accounts [2] have similarly been uninformative because they were dramatically underpowered. So maybe the best use of the latter half of the paper is to rework it as a warning to others that even an impressive panel like this is often going to be largely uninformative. Maybe the authors think differently about this. That's fine, but it seems like then the paper would really need to incorporate more detailed arguments about why the results are informative and rule out plausible effect sizes and/or effect sizes that would have electoral consequences.

Overall, I like this effort and think that the first part is valuable, but the second part is just doesn't really advance our knowledge and may confuse people — as is.

We agree that the descriptive results are particularly compelling and are one of the major contributions from these data, which, to our knowledge, are the best data available to

examine exposure to foreign influence accounts in this high-profile disinformation campaign. The more descriptive results in Figures 1 thru 3 provide indirect evidence that the Russian campaign was unlikely to meaningfully affect behavior: exposure is highly concentrated among a small subset of user; is exceptionally low relative to that from politicians and news media; and is primarily concentrated among harder-core Republicans. Our overall aim in estimating the relationship between exposure to foreign influence accounts and voting is then a way to provide a more direct examination of any potential relationship between exposure to posts from Russian influence accounts and political behavior, of which we now provide a more cautious interpretation. Given the limitations of the data, as the reviewer notes, these results in themselves are not dispositive. Instead, as we see it, it is the totality of the evidence presented—in particular the clear evidence of low rates of exposure and the non-strategic distribution of that exposure (primarily concentrated among respondents likely to be Trump supporters ex-ante), coupled with what the reviewer notes is evidence of consistently small evidence of advertising effects—that gives the most credence to our interpretation of the limited scope and potential influence of the Russian interference campaign.

We also appreciate the reviewer's observations on possible instruments for exposure. We think this could be a subject of future investigation, but as the reviewer himself notes, choosing any instrument would push us down the usual path of attempting to justify the instrument—and there are no shortage of possible objections to any instrument we could choose. We now note in the revised manuscript that a causal inference would be contingent on believing that choosing to follow IRA trolls, or accounts that retweet IRA trolls, is independent of the likelihood of changing one's vote intention over the course of the campaign. While we find it unlikely that an unobserved characteristic causing people inclined to switch to Clinton over the course of the campaign would also make them more likely to consume information from IRA accounts, this is still possible. In the revised manuscript we have aimed to be both more transparent and cautious in our language about the fact that we are not making a strong causal claim from the regression results themselves.

Sincerely, Dean Eckles

Other comments:

- I think a lot of the references to the SI/appendices are off throughout. Like p. 9 says to look to Figure B8 for the share of exposures that are indirect, but I think this is actually Figure C10.

Thank you for pointing out the discrepancies between references to the Appendix and indices used therein. We have corrected the reference to Figure for exposures and attempted to reconcile all references throughout the text of the manuscript.

- I think it would be better to use a quasi-Poisson model than to do the log-transformation of the number of exposures when used as an outcome (what is happening with the 30% of people with 0s?).

We thank the reviewer for this suggestion. In the revised manuscript we have implemented the model using a quasi-Poisson model in addition to the original OLS (i.e. $\log(x + 1)$) model. We have chosen to leave in the main manuscript the OLS model, because readers will likely be more familiar with the model, and we have thus put the quasi-Poisson model results in SI Appendix C4. As the results in the Appendix show, the relationship between our variable of interest in this model (i.e. political partisanship) and exposure is similar and highly significant.

- Similarly, when the number of exposures is a predictor, perhaps it could make sense to estimate a model with both an indicator for any exposure and $\min(0, \log(\text{exposures}))$ rather than a single term for $\log(\text{exposure}+1)$, which I assume is what used, though everything just says ' $\ln(\text{exposures})$ '.

This is an interesting suggestion, and one that we hadn't originally considered. The reviewer is correct that we use $\log(\text{exposure} + 1)$ as our variable of interest, which we have sought to make clearer in the revised manuscript through the text and figures. To address the reviewer's suggestion, we have supplemented the models that use $\log(\text{exposure} + 1)$ as a predictor by re-specifying these models to include both a binary variable to indicate whether a user was exposed to any tweets from the Russian foreign influence campaign and a variable defined as $\min(0, \log(\text{exposure}))$. We now include in SI Appendix H results for four different codings of the variable of interest: (a) $\log(\text{exposure} + 1)$, (b) a binary variable indicating exposure greater than 0, (c) a binary variable indicating whether a user followed a Russian influence account, and (d) a binary variable indicating exposure greater than 0 and $\min(0, \log(\text{exposure}))$. Because using both a binary variable and $\min(0, \log(\text{exposure}))$ may excessively complicate the currently neat presentation of our results in the main paper, we present a model with $\log(\text{exposure} + 1)$ and a model with a binary variable in the main manuscript.

- What's the reason for focusing on change scores (pre-post differences) rather than using regression adjustment? The former is a special case of the latter with the slope(s) fixed to 1, which is rarely the optimal choice. (Incidentally, this also creates the confusion I experienced about the standardized effect sizes, though I would also suggest just removing or substantially deemphasizing the standardized effect sizes.) As noted above, a fixed-effects approach could also make sense, as could other methods to more flexibly control for baseline behaviors and interests (Eckles & Bakshy 2020), though perhaps the sample sizes are again too small here.

The reason we chose to use change scores rather than regression adjustment is because in our case, with concerns about confounding (as the reviewer has also noted), there is evidence that they produce less biased estimates when the threat of confounding is larger. On this, we were guided by this simulation study (DeclareDesign, 2019) by the authors of Blair et al. (2019), who show that in the presence of confounding, change scores outperform models that place the pre-treatment measure on the right-hand side). We did fit regression models for the issue positions, polarization, and vote choice outcomes using the Wave 1 measure of each outcome as a right-hand-side variable, however, with substantively similar results, which we can include in the Appendix if the reviewer thinks it would be a useful addition.

- “As Panel A of Figure 1 shows further, posts from the Russian foreign influence campaign were by far the most prevalent, representing 86% of all exposures in respondents' timelines in the lead-up to the election.” (p. 8) Could be made more clear by noting this is 86% of the foreign influence campaigns included here.

It's true that this could be clearer. We have thus noted the fact that the 86% figure refers specifically to the influence campaigns from which we have data and which are used in the manuscript.

- Are self-reports of turnout reliable enough for the purposes they are put to here?

This is a useful question. It is known as a general matter that people tend to over-report voting. However, it is especially difficult and costly to validate voting records. The longest running national election study (the American National Election Study), for example, does not validate voting in its sample of respondents. In our revision, we have noted in the manuscript the fact that voting is self-reported.

Reviewer 3 comments

The manuscript presents a well-designed study on an important topic. The wide-ranging speculations about the impact of Russian interference on the 2016 elections have so far been backed by little direct evidence. The authors make a good case for the minimal effects of the interference campaign, at least as far as changing the minds of Twitter users is concerned.

We thank the reviewer for the supportive comments on the manuscript and constructive feedback on our empirical analysis.

I only have a couple of more significant concerns to mention about the methods and conclusions of the paper. The first one of these has to do with the main dependent variable, exposure to IRA misinformation. While the study consistently speaks of “exposure”, the actual measure used should more accurately be named “potential exposure” or “network exposure”. This may be good to explain early in the study as it does not become fully clear to the reader until the methods section.

We thank the reviewer for this suggestion, and we agree that the manuscript could be clearer that the data measure “potential exposures”. In the manuscript, we have followed the convention set by the most prominent similar study on fake news exposure (Grinberg et al. 2019), in which “exposures” are used explicitly as short-hand for “potential exposures”. This is because, as the reviewer suggests, data on “true exposures” are only available internally at social media companies (in our study, Twitter, although this problem is of course much wider), and thus “potential exposures” are researchers' best approximation to actual exposures. To the best of our knowledge, there is no evidence that measures of “potential exposures” as used by researchers are systematically biased in a direction that might concern exposure to Russian IRA accounts, but we recognize that this could indeed be the case. In the revised manuscript, to be clearer about this, at the point in the manuscript where we note our use of the term “exposures” as an economical shortcut for “potential exposures” (as done in, e.g. Grinberg et al. 2019), we now also further highlight the fact that because the data are “potential exposures” this should be viewed as a limitation of these data (see first paragraph of “Data and research design”).

The measure used here is based on the number of IRA-sourced tweets posted by accounts that the respondent follows. There are a couple of caveats to this metric: (1) A person is not guaranteed to have seen all tweets from their social network (or even to have had those tweets included in their stream). This would suggest many of the IRA tweets may not have been seen by the users; and (2) Twitter users often follow information beyond their immediate

networks, e.g. by following political hashtags. This would suggest that users who are considered “not exposed” could actually have seen IRA tweets.

This is all not to say that the metric used by the authors is not meaningful – it is more of a caution that it might be a noisy measure, even when aggregated over multiple months. I would suggest that the authors explicitly discuss this as a limitation of the study, as it could plausibly be one cause for finding null effects from misinformation exposure.

We thank the reviewer for raising these issues, and we agree that this is a potential threat to our inferences. As noted above, these are measurement limitations that are especially difficult to avoid in practice: Twitter may have these data, but researchers outside the company do not. Twitter introduced its algorithmic timeline some time in early 2016, which was a less complex and intrusive version of the timeline as that currently implemented. This potentially makes this measurement problem somewhat less problematic than if the data were more recent. In the revised manuscript, we have noted this point about measurement, adding this as a limitation to our results in the ‘Data and research design’ section of the main text of the manuscript.

Relatedly, I would suggest one small tweak in the regression models presented in the paper. That would be to include as a control not only “social media use”, but also the overall volume of tweets from the user's network. The reasoning is that a large volume of tweets from followed accounts reduces the probability of seeing any individual tweet. I also did not see a description of how “social media use” is measured or how it was distributed among respondents – I may have missed it but if not, that would be good to include.

We thank the reviewer for the suggestion, and agree that there are differences across respondents in their probability of actually seeing a tweet in their timeline. As clarification, our current approach is to control for the total amount of time respondents spend on social media (but we agree that a respondent who follows many accounts might have a lower probability of seeing a specific tweet from the accounts they follow). The variable that we use in our models is self-reported social media use, which is respondents' response to the question “How often do you use the Internet ... to use social media (Twitter, Facebook, Google+, etc.)?” (responses are measured on a Likert scale from ‘several times a day’ to ‘never’.) We agree that looking at the volume of tweets in respondents' timelines could control for an alternative/additional and, perhaps, more granular measure of exposure. We have thus included results for predicting exposure, issues positions/polarization, and voting behavior controlling for the number of tweets in a user's timeline (and for a finer-grained measure of racial groups as the reviewer also suggests in a comment further below) in Appendices C, E and F, with similar results to those in the main manuscript.

Somewhat surprisingly, the study never discusses user interactions with misinformation: e.g. the number of times people in the sample favorited, retweeted, or posted IRA misinformation materials. Based on previous research, it is likely that only a small percent of the respondents ever did that. It would nonetheless be useful to report it. As these behaviors send a very strong signal of interaction with misinformation, it would also be good to examine within-subject changes for those respondents in particular.

We thank the reviewer for this suggestion, and we agree that this is something that would be interesting to investigate. We do not possess the data required to carry it out. We collected the tweets of all friends followed by the respondents and matched those to tweets identified by Twitter as linked to Russia's foreign influence campaign. But while we do have the information on the overall level of engagement with these tweets from all Twitter users, this is not disaggregated by reactions of individual users in our data.

A couple of questions I have about the study revolve around the regression models examining potential effects from IRA misinformation exposure.

First, previous research has shown that political misinformation in 2016/2020 has heterogeneous effects on consumer attitudes depending on ideology. Exposure to the same misinformation item can shift the attitudes of strong Republicans in one direction, and that of strong Democrats in the opposite direction. Thus, it is plausible that the effect for the full sample could appear null, while there are significant but heterogeneous effects of exposure within groups. I would suggest looking at those groups separately to rule this out.

We agree with the reviewer that it would be interesting to examine heterogeneity in the potential relationship between exposure and each of the outcomes. We are limited in terms of statistical power given the size of our dataset (so estimates are imprecise). However, as an exploratory check on this, we nevertheless ran regression models examining the interaction between party ID and exposure for each of the vote choice outcomes, each model of which showed no meaningful results (although this might be expected given issues of statistical power with this kind of sub-group analysis). Due to a lack of power for these models, we chose not to include these models in the Appendix. However, if the reviewer thinks this would be a worthwhile addition, we will of course add it in a further revision.

In addition, as the authors point out, the misinformation content itself may have served a combination of goals. The agency is said to have spread multiple versions of stories with diverging messages meant for different audiences. Not sure if there is any way plausible way to identify/label the type of content users

were exposed to (perhaps identifying tweets that contain prominent hashtags representing different political views, etc). If that is possible, it would be quite valuable and interesting.

We thank the reviewer for this suggestion. Although we cannot rule it out, given the absence of a significant association between political ideology and changes in voting behavior, we suspect this is unlikely. Practically, however, the main challenge in examining this is the unavailability of a straightforward measurement strategy to determine the political stances of IRA tweets (stance detection is exceptionally challenging to do persuasively with social media data). Considering the complexity of this type of classification and the potential imprecision of the results, pursuing this is likely to introduce further measurement error instead of helping to resolve the problem of enriching the analysis of potential relationships with exposure.

Also important, I would recommend reconsidering the race/ethnicity treatment in the study. The authors examine race using the “White” vs. “Non-White” categories. It would be better to disaggregate Non-Whites and look at specific groups. This is especially key given that Russian misinformation seems to have exploited racial sentiments in the US. As can also be seen in the hashtag word cloud included in the supplementary materials, the IRA is known to have produced a large number of messages related to the Black Lives Matter movement. With that in mind, looking at Black respondents separately seems indicated.

We agree that further disaggregation and subgroup analysis could be useful to examine empirically. As noted above, we are limited to a large degree by statistical power to the extent that the number of Black respondents in our sample is small, at 146 (with other non-white groups being even a smaller fraction of the sample). Nevertheless, in the revised manuscript, we examined whether Black respondents were more likely than whites (and other non-whites) to be exposed to posts from Russian foreign influence accounts. The coefficient is positive in the $\log(\text{exposure} + 1)$ model and negative in the quasi-poisson model (on the edge of significance in one of four models), and thus it is difficult to draw strong conclusions about this more generally. These models are included in Appendices C, E and F. This does not in itself rule out the possibility of differences in exposure among this sub-group. But although the panel data uniquely enable us to examine the scope and potential influence of the IRA campaign overall, sub-group analyses in these data are not well-powered to pick up on difference among small groups. At the same time, we fully agree with the reviewer that the answer to this question would be worthwhile to know if it were possible to precisely examine empirically with these data.

One final note about the comparison of IRA troll tweets with other types of content. I found that interesting and informative, but my takeaway differs somewhat from that of the authors. The study notes that tweets from U.S. politicians outweigh those from Russian trolls 9:1. To me, one troll tweet for every nine politician tweets seems quite high, especially given that we are talking about only one of the bad actors out there vs. the entire American political elite. One thing that would be an interesting point of comparison to offer the readers is a chart showing the concentration of exposure to U.S. politicians (similar to Fig.1 B and C). That could help us better understand how the direct reach of legitimate political actors may compare to that of foreign misinformation sources.

We thank the reviewer for this excellent suggestion. A concentration plot for legitimate actors, such as media organizations and politicians, would indeed help illuminate the differences in outreach between these actors and state-backed electoral interference efforts that are investigated in the manuscript. In the revised manuscript we have added the requested plots to Appendix D, and we reproduce the figure below. The revised manuscript also notes that concentration of exposure to foreign influence accounts is especially pronounced, more so than that from politicians (and news media): analogous concentration plots for politicians and news media show that 1% of respondents account for 24% and 37% of exposures to posts from news media organizations and politicians respectively. This compares with 1% of respondents accounting for 70% of exposures to Russian foreign influence accounts. Similarly, 10% of respondents accounts for 67% and 79% of exposures to posts from news media and politicians, compared to effectively all (98%) of exposure to IRA accounts. In sum, exposure in the US to news media and political accounts is relatively high, but exposure to Russian IRA accounts is especially concentrated among an exceptionally small subset of users.

We also agree that the 9:1 ratio of politicians' to Russian IRA tweets in some sense appears non-trivial, it is useful to consider these numbers both in the context of the number of accounts that were linked to Russia-backed electoral interference campaign and the volume of tweets those generated. In our set of accounts we analyze the activities of 3,251 active malicious accounts that together were responsible for 7,427,397 tweets. At the same time we count approximately half the number of accounts as political elites (1,690) (and users generally follow many other users who are politically engaged). As can be seen in Panel D of the plot above, the general Pareto principle holds here as well, with 10% of politicians accounting for 90% of exposures. However, this is a dramatic difference to the picture in Figure 1C of the manuscript, where 1% of Russian troll accounts are responsible for 89% of exposures.

Reviewer 4 comments

This is an interesting and important paper which provides an analysis of the characteristics of U.S. twitter users who may have seen posts from the Russian interference efforts in the 2016 election. Broadly speaking the authors find that (1) the vast majority of potential exposures were to a small minority of accounts, (2) strong Republicans were the ones most likely to be following accounts that posted or reposted the material, and (3) that accounts which conceivably would have seen more exposure to posts were not owned by people experiencing substantial opinion shifts between April and October 2016.

We thank R4 for summarizing our findings, and for their supportive comments.

I quite like this paper and would want to see it published. It is at its best when it is presenting descriptive findings and providing interpretation. My suggestion is primarily on how to maintain that focus on description and avoid overclaiming evidence for causal effects. I split comments into a few thematic sections below.

(1) Descriptive Evidence and Claims

The paper makes a series of clearly descriptive claims (e.g. Figure 3, Panel A depicting disproportionate exposure of posts to Republicans) and a series of claims about 'effects' (Figure 3, Panel B showing a regression of exposure on party ID and 'controls'). These latter elements are not as well motivated as the descriptive claim and sit in an uncomfortable space between causal inference and description. For example, the panel B regression is motivated with the statement "To examine whether the findings in Panel A are an artifact of factors other than partisanship, we fit an OLS regression model to predict the (log) number of posts from Russian trolls that were in the timelines of each US respondent during the 2016 election campaign." Here I think the authors should make the claim and the relevance of this conditional expectation more explicit. I take the claim to be that they believe, somewhere, in some room, there was a conversation where there was a decision to target Republicans. They (implicitly) argue that an observable implication of that would be not just that the marginal distribution was more strongly tilted republican, but also that across demographic subgroups, republicans consistently saw more exposure. Having a small sample, they approximate these subgroups with a regression and then average over them. I don't think you have to put all of that on the page, but I do think if this is the train of thought more of it should be represented. As it is now, it seems like a 'regression that controls for things' because that is what is expected.

We thank the reviewer for this suggestion. As a result, in the revised version we have modified the description of Figure 3B to ensure that the coefficients displayed in Panel B are not interpreted as causal effects.

To the same end I think more humility is necessary in the claims around effects of exposure. Again, the authors implicit argument would seem to be that if exposure to this material caused a shift in opinions we would see that people with higher exposures did not move more in their opinions than than people with low exposures. (Note: I think this is what is going on but if I am correct there is an error in the caption of figure 4 which describes each as a model of exposure on ideology rather than the other way around. This is not helped by the fact that a similar style of plot early has independent variables down the rows rather than outcomes). This is a helpful descriptive implication of the authors purported state of the world, but it should be clearly stated as such rather than presented implicitly as an estimate of a causal effect.

We thank the reviewer for raising these issues and we have sought to address them in several ways. First, we added a y-axis label to Figure 3B to explicitly note that each variable is a predictor (not an outcome). More importantly, in Figure 4 we added a label noting that each row displays a separate model and that the outcome variable is conveyed in each row. Second, the caption of Figure 4 now refers to estimated associations (between exposures and changes in ideology or issue positions). Finally, in response to the reviewer and the concerns raised by other reviewers, we have modified the language used throughout the manuscript to emphasize the descriptive nature of our analyses, and the limitations in the data and voting behavior regression models.

This is all to say that the authors are making claims in a difficult situation where good data is scarce, but that's all the more reason to make the claims humbly and with a clear presentation of the inferential jumps that are required to believe the case. To the extent possible I would also be careful about how these findings are presented in the abstract as we know in practice many people won't get further than that.

In the revised manuscript, we have edited the language in the abstract to be more cautious about how our conclusions are presented (for example, removing references to 'effects', which are now heavily caveated within the manuscript generally), to clarify that our evidence is primarily descriptive. In addition, the revised manuscript is also more careful in describing our findings about the relationships between exposure and attitudes and voting behavior.

(2) Measurement Error and Null Findings

The authors are in the unenviable position of arguing for null findings in a setting with a truly astonishing amount of measurement error. Ostensibly the construct is not whether a post from an account appeared in a timeline, but whether the individual actually read or engaged with the post. Because the measurement error of this construct isn't classical, it is hard to know in what way it biases results, but I think it is reasonable to suppose that because the measure is almost certainly an overestimate of real engagement with these particular posts, it is plausible that these obscure any real effect on a small sub-population of users. In this sense it is very plausible that the measurement error leans in the direction of the author's finding which is an uncomfortable place to be. I'm not sure this is avoidable but it feels like it should be acknowledged.

I also applaud the authors for the equivalence testing but I think this section would be more readable and informative if the argument was made more substantively rather than using conventional fractions of standard deviations to think about negligible size.

We thank the reviewer for raising these concerns. Indeed, as outside researchers not embedded within a social media company we do not have access to data on direct engagement with posts encountered by respondents in their timelines. We have sought to acknowledge this data limitation.

Regarding equivalence testing, in the revised manuscript we have now changed the voting behavior results so that they are on a much more intuitive percentage point scale, because we agree that standard deviations for voting behavior obscures the magnitude of the relationships and uncertainty in estimation. To do this, we predict the mean change in voting by fitting the voting behavior models and then calculating the predicted change in vote choice by setting exposure for all respondents in the dataset to 0, and then setting exposure to the actual amount of exposure respondents received in the observed data. This provides an estimate of the relationship between exposure and voting in the election under the counterfactual that no respondent was exposed to any content from Russian foreign influence accounts (compared to actual exposure among those exposed). We use simulation from the variance-covariance matrix of the models to capture uncertainty in these estimated relationships. In the revised manuscript, we show these results graphically in Figure 6 in the revised manuscript, and have reproduced them below, where the vertical red lines indicate the median prediction, and vertical black lines indicate 90% prediction intervals.

For the relationship between exposure and changes in issues positions and polarization, the manuscript continues to use standard deviations because the original scale itself is wholly arbitrary (0-100 issues scales), and thus standard deviations are likely the most intuitive measure for readers.

(3) Bringing Caveats Forward A number of very reasonable caveats and limitations to the work are mentioned in the final paragraph of the paper. I think this comes too late. The authors begin the final paragraph “Despite these findings, it would be a mistake to conclude that simply because Russian IRA trolls activity on Twitter did not meaningfully impact individual-level attitudes that other aspects of the Russian foreign influence campaign did not have any impact on the election or faith in American electoral integrity.” But whether intended to or not, this is the opposite of the impression conveyed throughout the entire rest of the paper. Even this statement I think overstates the evidence available for identifying the ‘meaningful impact’ on attitudes. I think these points need to be raised earlier.

This work is important- even with the caveats stated clearly and prominently earlier in the paper. At least for my part, given the limitations of the argument more prominent would make the paper more credible, not weaker.

We appreciate this suggestion, and we have added the important caveat about indirect effects of state-sponsored operation to the ‘Introduction’ (second paragraph). More generally, we are more cautious in our language throughout about the limitations of the data and the potential (non-)relationship between exposure and attitudes and voting behavior.

(4) Smaller Details

(a) when presenting the survey the paper says that “The composition of survey respondents is approximately representative of the demographic profile of the

US voting-age public” and does not further raise the issue. This strikes me as not obviously ideal given that the inference is about the Twitter population and not the U.S. population which is notably different. Perhaps it would be possible to discuss explicitly what the target population is?

(b) The authors write: “Finally, we use data released by Twitter to identify the posts in survey respondents' timelines” Can you provide additional details on exactly what this procedure is and what it does and does not capture? It would be very helpful to clarify what we think this translates to in terms of rates of people actually reading posts. My guess is that the authors don't necessarily know, but being more expert than the readers on the subject, even providing some guidance of plausible values would be helpful.

(c) I think Figure 2 panel B has a mislabeled Y-axis and should be median?

(d) The reference to appendix B3 in the text should (I think) be C3 as there is no B3 and C3 would seem to be the correct material.

(e) Appendix G (the software statement) is a wonderful contribution.

(a) We agree that the Twitter population would be an appropriate target. In SI Appendix A2 we report both how closely our sample approximates the US population, and the Twitter population (the latter being based on estimates from Pew Research Center).

(b) To clarify, only Twitter itself has data on who clicks on a story, such that we do not know whether respondents read posts (with links) that they are exposed to. Our measure of potential exposure is based on whether respondents followed particular accounts and thus would be exposed in their timelines to the tweets their friends send and retweet (e.g. whether those accounts they follow potentially retweeted IRA content into their timelines, or are IRA accounts themselves). Knowing whether respondents' read articles that IRA accounts tweeted would enrich our analysis, but we (nor others to our knowledge) unfortunately do not have access to those data.

(c) We thank the reviewer for this correction, and we have changed “Avg.” in Panel B of Figure 2 to “Median.”

(d) Thank you very much for flagging this. We have fixed this reference mismatch in the Appendix. We have also noticed other discrepancies between indices in the manuscript and Appendix and attempted to correct them through the text.

(e) Thank you for this comment. We believe that this is an important, if small way to give credit to the developers who make applied research possible.

Overall:

This paper is important, informative and likely to be hugely influential. While I think the authors make a compelling case with imperfect evidence, I also

think they misrepresent the strength of that evidence at times particularly for a lay reader. Being clearer about the limitations (and why stronger evidence will likely never be available) seems like it would only make the paper stronger.

We thank the reviewer again for the constructive feedback. We are glad that the reviewer appreciated our ambition to provide what we consider the best currently available evidence of variation in exposure to Russian foreign influence campaign during the 2016 election campaign, and its relationship with attitudes and voting behavior. Relatedly, in the 'Introduction' section we added that our study is the first to permit conclusions backed by empirical data from the election period about the relationship between these variables. We have sought to make it clear that our contribution lies in providing “a near best-case observational design” ('Data and research design' section of the main manuscript) despite the data limitations. When discussing our key independent variable, “potential exposures”, ('Data and research design' section) we note the limitations of such a measure and in paragraphs below that describe why this is the best practicable measure given our research question. While we would prefer to make stronger claims, we believe these are the most important and meaningful conclusions that are still defensible with our data and research design.

References

Blair, Graeme, Jasper Cooper, Alexander Coppock and Macartan Humphreys. 2019. "Declaring and Diagnosing Research Designs." *American Political Science Review* 113(3):838–859.

DeclareDesign. "Use change scores or control for pre-treatment Depends on the true data generating process." DeclareDesign, January <https://declaredesign.org/blog/use-change-scores-or-control-for-pre-treatment-outcomes-depends-on-the-true-data-g.html>.

REVIEWERS' COMMENTS

Reviewer #1 (Remarks to the Author):

Referee Report for "Exposure to the Russian foreign influence campaign on Twitter in the 2016 US Election and its relationship to political attitudes and voting behavior" (302261_1)

In this revision, the authors have improved an already strong manuscript and I like their responses to my comments. My only remaining quibble is that the causal estimand is still not explicitly defined, despite the language on page 17 saying that the *estimates* are "driven by those (heavily) exposed to messages from these accounts." That's of course true -- but it would good to clarify for readers what the estimand is (which is an ATT, so far as I can tell) separately from what the estimator happens to shoot at.

I thank the authors for their excellent work and look forward to seeing this paper in print.

Reviewer #2 (Remarks to the Author):

I remain enthusiastic about this paper, especially the analysis of who was exposed to content from known IRA accounts on Twitter in the 2016 US election.

I really appreciated the additional comparison with effects on vote choice, even if I don't reach the same broad conclusions as the authors. I was a bit disappointed there wasn't more comparison of the estimates for the issue attitudes and other literature. I think a reader will still not really be clear whether the kinds of effects seen elsewhere are ruled out or not.

When it comes to informative estimates of effects on vote choice, I don't see Broockman & Green (2014) as helpful. This is an obviously underpowered study of digital ads that produces null estimates that, if I understand them correctly, are compatible with those ads being the most cost-effective campaign media expenditures possible (and maybe even more cost-effective than personal contact).

The authors then write:

"Estimates from these distinctly targeted advertisements might be thought of as an upper bound on any theoretically achievable relationship from exposure to the Russian foreign influence campaign." Doesn't this then say that we shouldn't really update our beliefs about effects of exposure to IRA content much at all? That is, the confidence intervals we have here include what the authors describe as "an upper bound on any theoretically achievable" effect, and we can't rule out effects of that size. Again, this points to the idea that (at least for political scientists) the analysis of exposure effects (even if we believe the causal identification) doesn't rule out most plausible effect sizes.

Thus, here's how I might summarize the empirical results (first assuming we believe the causal identification): The effects of exposure to a single typical post from the IRA are not much larger than consensus estimates of the effect of exposure to a video campaign ad.

Overall, I continue to think that the analysis of predictors of exposure is valuable and should be published. I don't really find the analysis of exposure that helpful, but I think the current version is at least more clear about the remaining uncertainty, so I guess I have less strong objections to it now.

Other comments:

- "This result is similar for polarization. Of the two statistically significant coefficients across all models (representing 6% of our coefficients, as would be expected by chance), neither is in a direction that would suggest that exposure to posts from Russian foreign influence accounts is related to an increase in perceived polarization."

I wondered whether, given how this is measured here, whether this is the ostensible aim of these efforts. For example, we might think this content could aim to make the parties seem indistinguishable (this has been posited as a goal of some of the IRA content documented on Facebook) or to make Trump look moderate.

- I appreciated being able to look at the regression tables (e.g. Table E10). This helped me think (somewhat informally) about how important other variables were as predictors of attitude change. For example, it seems like often the coefficients for most demographics are not so large either. One way of summarizing this is that none of the predictors here seem to do a very good job of predicting whatever attitude change is present in the data.

Sincerely,
Dean Eckles

Reviewer #3 (Remarks to the Author):

The authors have reasonably addressed my feedback on their manuscript in the revised version. I agree with other reviewers that the descriptive portion of the study is more compelling than the vote choice examination, especially given limitations of the data. Reading the revised manuscript, it seems to me the authors have sufficiently softened causal claims and discussed important caveats. I would thus support publishing the study in its current form.

Reviewer #4 (Remarks to the Author):

It is wonderful to see this paper again and it was a pleasure to read the thoughts of the other reviewers. Overall, I think my concerns were generally well-addressed and I'm happy to support publication.

As the authors themselves note, none of the evidence here is by itself determinative, but they are admirably clear about the broader triangulation strategy.

I have two small notes:

(1) The new figure 6 on simulated vote change is quite welcome. I think the framing of these results again slides into the causal direction. On page 22, the authors write "Estimates from these distinctly targeted advertisements might be thought of as an upper bound on any theoretically achievable relationship from exposure to the Russian foreign influence campaign" but the immediate next sentence (starting a new paragraph) says " Given that the data are observational, we stress that the relationships that we estimate cannot be confidently said to be causal." These two things feel in contrast with each other to me. It is perhaps a personal stylistic preference, but it feels like if you want to make the claim, it is better to say that you want a particular causal effect, it would be identified under these assumptions, and then acknowledge the assumptions aren't really met (perhaps with a sensitivity analysis of the assumptions). Regardless, what the paper is doing here doesn't seem too far outside the norm.

(2) "log(Exposure)" in the titles of the left-side plots in Figure 4, should (I think!) be changed to "log(Exposure + 1)" (see also Figure 4 caption and title in Figure 5). If you are making changes to that figure, I'd contemplate adding some visual signal of positive direction indicating favorability to Trump and negative indicating favorability to Clinton.

Reviewer 1

*In this revision, the authors have improved an already strong manuscript and I like their responses to my comments. My only remaining quibble is that the causal estimand is still not explicitly defined, despite the language on page 17 saying that the *estimates* are “driven by those (heavily) exposed to messages from these accounts.” That’s of course true – but it would good to clarify for readers what the estimand is (which is an ATT, so far as I can tell) separately from what the estimator happens to shoot at.*

I thank the authors for their excellent work and look forward to seeing this paper in print.

We thank the reviewer for this request to specify the target estimand more explicitly. In the paragraph to which the reviewer refers, we have added, when discussing our model setup, that “If exposure were unconfounded between survey waves (a strong assumption), the estimand in this model would be the average treatment effect on the treated to the extent that levels of exposure is as observed in the data.”

Reviewer 2

I remain enthusiastic about this paper, especially the analysis of who was exposed to content from known IRA accounts on Twitter in the 2016 US election.

I really appreciated the additional comparison with effects on vote choice, even if I don't reach the same broad conclusions as the authors. I was a bit disappointed there wasn't more comparison of the estimates for the issue attitudes and other literature. I think a reader will still not really be clear whether the kinds of effects seen elsewhere are ruled out or not.

When it comes to informative estimates of effects on vote choice, I don't see Broockman & Green (2014) as helpful. This is an obviously underpowered study of digital ads that produces null estimates that, if I understand them correctly, are compatible with those ads being the most cost-effective campaign media expenditures possible (and maybe even more cost-effective than personal contact).

The authors then write:

“Estimates from these distinctly targeted advertisements might be thought of as an upper bound on any theoretically achievable relationship from exposure to the Russian foreign influence campaign.” Doesn't this then say that we shouldn't really update our beliefs about effects of exposure to IRA content much at all? That is, the confidence intervals we have here include what the authors describe as “an upper bound on any theoretically achievable” effect, and we can't rule out effects of that size. Again, this points to the idea that (at least for political scientists) the analysis of exposure effects (even if we believe the causal identification) doesn't rule out most plausible effect sizes.

Thus, here's how I might summarize the empirical results (first assuming we believe the causal identification): The effects of exposure to a single typical post from the IRA are not much larger than consensus estimates of the effect of exposure to a video campaign ad.

Overall, I continue to think that the analysis of predictors of exposure is valuable and should be published. I don't really find the analysis of exposure that helpful, but I think the current version is at least more clear about the remaining uncertainty, so I guess I have less strong objections to it now.

We agree that a comparison to the literature concerning the effects on issue attitudes would be useful. This is somewhat challenging to the extent that there is relatively little field experimental research on media campaigns and their effects on attitudes (as

compared to that on, for example, vote choice or turnout). In the revised manuscript, we now compare our estimates to those from Kalla and Broockman (2022), who estimate the effect of a series of media campaigns on political attitudes toward immigration and LGBTQ issues, and find effectively null effects on policy attitudes. We state this in the revised manuscript as follows:

“For comparison, the absence of any relationships between exposure and changes in issue positions is consistent with recent large-scale field experimental research in the US that finds near-zero effects ($\beta \approx 0.01$ SD) of exposure to targeted issue advertisements on changes in issue positions (LGBTQ and immigration policy preferences) (Kalla and Broockman, 2022).”

In the revised manuscript, we also now frame the results in a similar fashion to that suggested by the reviewer, as follows:

“And, put differently, the minimal and non-significant relationships that we observe between exposure to posts from disinformation accounts and voting behavior are similar in magnitude to the minimal relationships observed in research on the effects of traditional offline and online campaigns (Nickerson and Rogers, 2020).”

Other comments:

- *“This result is similar for polarization. Of the two statistically significant coefficients across all models (representing 6% of our coefficients, as would be expected by chance), neither is in a direction that would suggest that exposure to posts from Russian foreign influence accounts is related to an increase in perceived polarization.” I wondered whether, given how this is measured here, whether this is the ostensible aim of these efforts. For example, we might think this content could aim to make the parties seem indistinguishable (this has been posited as a goal of some of the IRA content documented on Facebook) or to make Trump look moderate.*

We have searched for, but have been unable to find research that posits that the goal of the Russian Internet Research Agency was to increase perceptions that Hilary Clinton and Donald Trump were ideologically similar. To the best of our knowledge, previous assessments of Internet Research Agency efforts (DiResta et al., 2019) do not point to such a subtle strategy employed by the Russian foreign influence campaign in the 2016 US Election.

Empirically, the results show, generally, that there is neither an increase nor decrease in perceptions of polarization among those exposed to content from the Russian foreign influence campaign. We would be happy to include any citations to research that suggests that the aim of the campaign may also have been to decrease these perceptions (empirically we do not find this, however).

- I appreciated being able to look at the regression tables (e.g. Table E10). This helped me think (somewhat informally) about how important other variables were as predictors of attitude change. For example, it seems like often the coefficients for most demographics are not so large either. One way of summarizing this is that none of the predictors here seem to do a very good job of predicting whatever attitude change is present in the data.

Sincerely, Dean Eckles

We thank the reviewer for pointing out this relationship between our control variables and outcomes. To address this, we have added a note, which reads: “It is also worth noting that none of the other explanatory variables (with the exception of sex in some models) used as controls appear to be statistically significant predictors of the change in voting preferences (see SI Appendix E1, E2, E3).”

Reviewer 3

The authors have reasonably addressed my feedback on their manuscript in the revised version. I agree with other reviewers that the descriptive portion of the study is more compelling than the vote choice examination, especially given limitations of the data. Reading the revised manuscript, it seems to me the authors have sufficiently softened causal claims and discussed important caveats. I would thus support publishing the study in its current form.

We thank the reviewer for the constructive feedback and appreciate their support for the manuscript.

Reviewer 4

It is wonderful to see this paper again and it was a pleasure to read the thoughts of the other reviewers. Overall, I think my concerns were generally well-addressed and I'm happy to support publication.

As the authors themselves note, none of the evidence here is by itself determinative, but they are admirably clear about the broader triangulation strategy.

I have two small notes:

(1) The new figure 6 on simulated vote change is quite welcome. I think the framing of these results again slides into the causal direction. On page 22, the authors write “Estimates from these distinctly targeted advertisements might be thought of as an upper bound on any theoretically achievable relationship from

exposure to the Russian foreign influence campaign” but the immediate next sentence (starting a new paragraph) says “Given that the data are observational, we stress that the relationships that we estimate cannot be confidently said to be causal.” These two things feel in contrast with each other to me. It is perhaps a personal stylistic preference, but it feels like if you want to make the claim, it is better to say that you want a particular causal effect, it would be identified under these assumptions, and then acknowledge the assumptions aren’t really met (perhaps with a sensitivity analysis of the assumptions). Regardless, what the paper is doing here doesn’t seem too far outside the norm.

In this section, the estimates which the manuscript refers to as reasonable upper bounds are causal estimates of the effect of targeted advertisements in a few high-quality recent studies Kalla and Broockman (2018); Broockman and Green (2014); Coppock, Hill and Vavreck (2020). This addition was included because a reviewer suggested that we note the magnitude of the size of field experimental effects as a rough basis for comparison to the relationships we observe in the case of exposure to Russian foreign influence accounts. We nevertheless thought it best to emphasize after noting this that our data, unlike the cited studies, are observational to avoid any unwarranted allusion that the data are experimental.

(2) “log(Exposure)” in the titles of the left-side plots in Figure 4, should (I think!) be changed to “log(Exposure + 1)” (see also Figure 4 caption and title in Figure 5). If you are making changes to that figure, I’d contemplate adding some visual signal of positive direction indicating favorability to Trump and negative indicating favorability to Clinton.

The reviewer is correct about this inaccuracy. We have thus modified the title in the relevant panel to read “log(Exposure + 1)”.

References

- Broockman, David E. and Donald P. Green. 2014. “Do Online Advertisements Increase Political Candidates’ Name Recognition or Favorability? Evidence from Randomized Field Experiments.” *Political Behavior* 36(2):263–289.
- Coppock, Alexander, Seth J. Hill and Lynn Vavreck. 2020. “The Small Effects of Political Advertising are Small Regardless of Context, Message, Sender, or Receiver: Evidence from 59 Real-time Randomized Experiments.” *Science Advances* 6:1–6.
- DiResta, Renee, Kris Shaffer, Becky Ruppel, David Sullivan, Robert Matney, Ryan Fox, Jonathan Albright and Ben Johnson. 2019. *The Tactics & Tropes of the Internet Research Agency*. Technical report New Knowledge.
- Kalla, Joshua L and David E Broockman. 2018. “The Minimal Persuasive Effects of Campaign Contact in General Elections: Evidence from 49 Field Experiments.” *American Political Science Review* 112(1):148–166.
- Kalla, Joshua L. and David E. Broockman. 2022. ““Outside Lobbying” Over the Airwaves: A Randomized Field Experiment on Televised Issue Ads.” *American Political Science Review* 116(3):1126–1132.
- Nickerson, David W. and Todd Rogers. 2020. “Campaigns Influence Election Outcomes Less than You Think.” *Science* 369(6508):1181–1182.